# Private investments in climate change adaptation are increasing in Europe, although sectoral differences remain
Ignasi Cortés Arbués [1] ✉, Theodoros Chatzivasileiadis [1], Servaas Storm[2], Olga Ivanova [3] & Tatiana Filatova [1] ✉

Climate-induced hazards are becoming more frequent and severe, causing escalating economic losses worldwide. Consequently, climate change adaptation is increasingly necessary to protect people, nature and the economy. However, little is known about who is adapting and how much they spend on adaptation measures, especially in the private sector. This article focuses on firms—the backbone of economic development, yet understudied in climate adaptation research. Here we present insights from a unique panel dataset detailing businesses' adaptation investments across 28 European countries (2018–2022), 5 hazard types, and 19 economic sectors. Our descriptive analysis reveals low but increasing adaptation investments across Europe (0.15–0.92% of national gross domestic product, annually increasing by 30.6–37.4%). Moreover, we highlight considerable differences in adaptation intensity across sectors, including low adaptation intensity in manufacturing and retail trade. Additionally, our econometric analysis indicates that public adaptation spending crowds in private investments in adaptation, highlighting opportunities to facilitate autonomous adaptation.

The environmental and societal impacts of climate change are already visible worldwide[1], and are expected to intensify given current greenhouse gas emissions' trajectories[2]. Specifically, extreme weather events are becoming more frequent and severe[2–5], sharply increasing climate-induced economic damages[6]. Only within the European Union (EU), economic losses associated with extreme weather events amounted to 59.4 billion EUR in 2021, and 52.3 billion EUR in 2022[7]. The societal consequences of these adversities become increasingly painful, as exemplified by extreme heat across Asia[8], catastrophic hurricanes in the USA[9], and devastating floods in multiple African river basins in 2024[10]. Even in a relatively well-protected Europe, catastrophic flooding across Central Europe in September[11] and in Spain in October[12] jointly killed more than 200 people and forced mass evacuations and havoc. In this context, climate change mitigation alone is insufficient to curb losses of lives and livelihoods[13]. Climate change adaptation (CCA) is crucial to protect societies in the coming decades against already committed climate change and to facilitate economic development. To reduce the burden on future generations, CCA must be planned and adopted today[14].

However, the development and implementation of CCA measures is complex, as they involve both public and private actors[15] with different adaptation priorities. Specifically, private firms are expected to adapt according to self-interest[16]—i.e., protecting their own assets while minimising their costs, whereas governments have broader priorities that include protecting citizens and building critical infrastructure[17]. Yet, as economic damages caused by extreme events continue growing[6], both governments and firms themselves have strong incentives to protect business activity. A climate-resilient private sector is essential to guarantee economic development[2], provide jobs, and protect tax revenues that fund public CCA[18]. Still, a 2022 study conducted by S&P Global shows that only one in five surveyed companies had a physical risk adaptation plan[19]. Consequently, studying private-sector CCA and its dynamics are two key priorities for adaptation research and policy[20].

Nevertheless, while public CCA policies have been extensively researched and are increasingly included in macroeconomic assessments[13,18,21,22], private CCA is comparatively understudied. In fact, the private CCA literature largely focuses on measures taken by households[20,23–25] rather than by firms[26,27]. This is not surprising, as besides those compiled by the CDP[28], there are few publicly available data on adaptation measures taken by businesses, which are hesitant to share their investment plans for competitive reasons[16]. While there are no perfect

[1]Department of Multi-Actor Systems, Faculty of Technology, Policy and Management, Delft University of Technology, Delft, The Netherlands. [2]Department of Values, Technology and Innovation, Faculty of Technology, Policy and Management, Delft University of Technology, Delft, The Netherlands. [3]PBL Netherlands Environmental Assessment Agency, The Hague, The Netherlands. ✉e-mail: i.cortesarbues@tudelft.nl; t.filatova@tudelft.nl

datasets, their unavailability in the domain of private CCA is exceptional. As a result, quantitative analyses of private-sector CCA have so far been scarce, with hardly any macro-level impact assessments explicitly accounting for CCA measures implemented by firms. Previous studies tend to focus on the economic micro-level, at the scale of cities or regions, often assessing individual economic sectors, like manufacturing[29] or agriculture[30,31]. These analyses are often case- and hazard-specific, such as a survey of small- and medium-sized manufacturing firms in Ho Chi Minh City focused on unveiling business characteristics that determine whether a firm adapts to flooding[29]. Hence, while there is useful evidence of what drives private CCA in specific regional and sectoral contexts, our understanding of how much businesses actually invest in CCA and what these patterns look like across economic sectors and countries remains limited.

Accordingly, we identify three core challenges facing private-sector CCA research. Firstly, the economic impacts of different climate-induced hazards vary by location, which could trigger higher or lower CCA investments. For instance, coastal regions like the Veneto (Italy) are susceptible to sea-level rise (SLR)[32], while arid regions in Greece, Spain or Portugal are vulnerable to wildfires[33]. Clearly, within Europe, uneven geography and climate affect the adaptation needs of various regions, emphasising the importance of economic assessments that encompass CCA for different climate hazards. Secondly, the channels by which different hazards affect economic activity vary, with certain sectors more vulnerable to specific hazards than others[34]. For instance, capital-intensive sectors like manufacturing are particularly vulnerable to floods, as their machinery gets physically damaged[29]. Conversely, the impact of heatwaves on the workforce is more widespread, as labour productivity losses affect most economic sectors[35]. Clearly, CCA can vary along multiple dimensions—hazard type, sector, geographical region—and this heterogeneity warrants further study at both micro- and macro-levels.

Finally, potential synergies between investments in public and private CCA remain underexplored. Traditionally, CCA planning has relied on the public sector to supply and maintain critical infrastructure (e.g., coastal protection infrastructure to combat SLR[22,36]), while private CCA measures are more spatially localised, like installing air conditioning units to mitigate heatwave impacts[37]. In certain scenarios, public and private CCA measures have acted together (e.g., early-warning systems and dry-proofing during flooding events)[23], leading to an increasing interest in studying interactions between private (largely autonomous) adaptations and public (planned) adaptations[38]. In some contexts, publicly planned CCA measures have shown potential to both signal the need for private actors to adapt[39] and increase the capacity of small businesses to undertake autonomous adaptation[40] – including through public-private partnerships (PPP)[41]. However, the specific interplay between investments in CCA by firms and by governments is unclear. Public CCA may discourage private action[42], while too much CCA could lead to sub-optimal decision-making, wasting valuable economic resources that could have been used for other purposes. Additionally, the allocation of these scarce public resources is often driven by benefit-cost maximisation, which can reinforce existing inequalities by prioritising protecting high-value assets owned by richer individuals[43]. Therefore, it is essential to understand the relationship between public and private CCA patterns and policies, as all economic actors will have to be involved to face upcoming climate threats.

In addressing these gaps, our article contributes to the CCA literature with two types of empirical analysis. First, we start with a quantitative overview of the current state of private adaptation through the lens of sectoral investment (henceforth, referred to simply as adaptation investment(s) or expenditure(s)). These include all capital expenditures on CCA, including those funded through foreign direct investments. Here, we present relevant adaptation expenditure patterns across economic sectors and hazard types for 28 European countries—the EU and the United Kingdom (UK)—between 2018 and 2022. Second, we proceed by identifying socioeconomic factors with statistical effects on private CCA investment, considering the heterogeneous nature of climate-induced hazards. Specifically, we examine the relationship between public and private CCA expenditure

across the 28 countries by conducting a panel regression analysis on the effects of various factors on private-sector CCA. To achieve these goals, we use a unique CCA investment dataset based on real adaptation-related transactions provided by kMatrix Data Services[44] (see Methods for an extensive description, and Supplementary Note 2 for a taxonomy of CCA measures considered), previously used to assess adaptation responses in global megacities[45]. The CCA investments are split in 28 countries, 19 economic sectors and five climate hazard types, over a 5-year period (2018–2022). By leveraging the dataset's dimensionality, we unveil how much European sectors are investing in CCA relative to their size in each national economy, and identify socioeconomic characteristics with significant effects on private CCA investment. Our analysis highlights that the European adaptation expenditures represent between 0.15–0.92% of national gross domestic product (GDP) and are increasing steadily (between 30.6–37.4% per year), although key economic sectors like manufacturing remain non-adaptation-intensive. Moreover, we find that public adaptation expenditure crowds in private adaptation investments and could help enable autonomous private CCA. We conclude by discussing the implications of our study for the European economy and beyond, and how synergies between public and private investments can help to reduce the adaptation gap[46].

## Results
### Dynamics of adaptation investments across countries
Investments in CCA by the private sector in Europe are small, though they have been increasing considerably in recent years. According to the dataset used throughout this study (see Methods), the total yearly investment in CCA within the EU and the UK was 15.4 billion EUR in 2018, up to 52.9 billion EUR by 2022 (in 2018 constant prices), a 243% increase over 5 years. However, these investments are not evenly distributed across the 28 countries, or even within them, as different sectors adapt at varying rates and focus on different climate-induced hazards. On the aggregate, the data show that, when normalising by a country's GDP, the share of adaptation expenditure has increased annually over the period 2018–2022 for all 28 countries (Fig. 1). The consistent increase in adaptation shares suggests that economic impacts of various global shocks (COVID-19 pandemic, geopolitical conflicts) have not affected the general investment trend. Overall, in 21 out of the 28 countries in the dataset, adaptation-to-GDP ratios lay between 0.15 and 0.25% when averaged over the five years, hinting at a relatively even share of adaptation investment across the EU and the UK. As expected, given its historic exposure to both river and coastal flooding[47], and high potential for compound hazards, the Netherlands spends the most on adaptation relative to its economy, averaging 0.58% and peaking at 0.92% in 2022, followed by Greece, Croatia, and Italy. On the opposite end of the spectrum, Ireland and Luxembourg trail in total adaptation investments, averaging below 0.1% of GDP over the 5-year span. Besides the Dutch outlier, there is a relative clustering of coastal nations in Southern, Central and Eastern Europe at the higher end of the adaptation spectrum, which is expected given the higher incidence of actual and projected climate damages[48–50]. Given their comparatively lower levels of income, similar adaptation measures might represent higher relative costs per unit of GDP. Conversely, the lower half of the ranking is populated by smaller and Northern European states, which are projected to be relatively less impacted by climate shocks[48–50]. In the long term, as extreme weather events become more frequent and intense, adaptation needs are likely to increase across Europe, placing a greater financing burden on countries in Southern, Central and Eastern Europe with higher projected climate damages and lower economic productivity[48].

### Economic sectors diverge on adaptation investment intensity
Private firms adapt differently across locations, as geography and climate expose them to extreme weather events at varying degrees. However, both the sector to which a business belongs[16] and firm size[29] can affect how much turnover is invested in adaptation. Our dataset aggregates all firms in a country into 19 NACE 2 economic sectors, such that little can be inferred

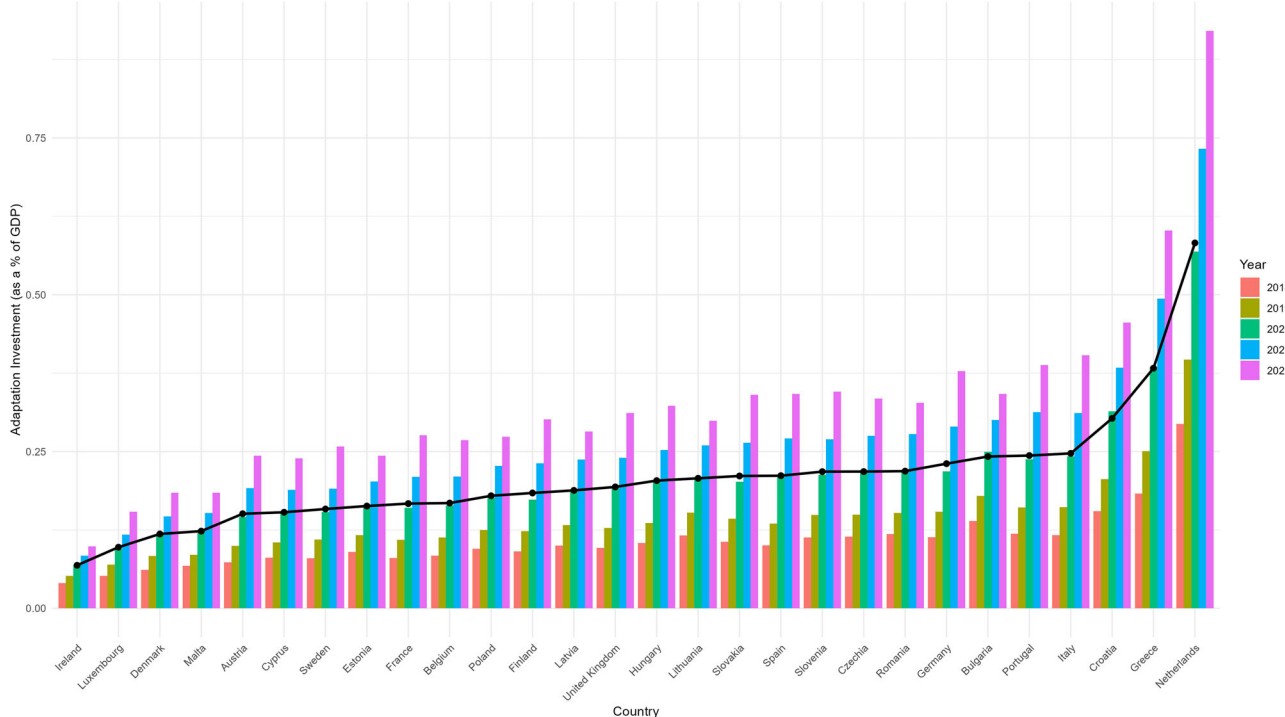

**Fig. 1 | Total national adaptation investment as a percentage of GDP for the EU countries and the UK, over the period 2018–2022.** The black dots represent the average total adaptation investment as a percentage of GDP over the five years for each country, and is used to sort them from lowest (Ireland, 0.07%) to highest (Netherlands, 0.58%). The graph can be interpreted as a ranking of the 28 countries in terms of adaptation expenditure after normalising by the size of their economies. The presented data have been aggregated over 19 NACE 2 economic sectors and five hazard categories.

about the impact of firm size on adaptation investment. Still, we can assess sector-level investments in each country to identify industry-specific trends across the EU and the UK (Fig. 2). We define a ratio between the sectoral share in total national adaptation expenditure and the sectoral share in GDP, averaged over the period 2018–2022. This intensity indicator can be interpreted as a measure of how much a sector is investing in adaptation relative to its contribution to the national economy.

Sectoral expenditure trends are remarkably consistent across countries (Fig. 2), allowing us to identify sector-level clusters of relative adaptation investment intensity. On the back end of the ranking, Manufacturing (C) and Wholesale and Retail Trade (G) invest, in most countries, four (0.25) to five (0.20) times less on adaptation compared to their sectors' gross value added (GVA) share in GDP. These results are important as Manufacturing is the largest contributor to GDP for most European countries, reaching 25% in some Central European countries, and over 30% in Ireland[51]. It is also a sector with substantial assets in buildings and machinery, which could be physically damaged by adverse climate impacts; these damages may hinder production and cause massive ripple effects through supply chains[29]. However, not all sectors feature such regionally-uniform expenditure trends, as we can observe in Accommodation and Food Services (I). As a sector closely linked to tourism, it holds great economic importance to countries in Southern Europe like Greece or Spain[51], and it is precisely in these countries where sectoral adaptation intensity is lower, despite high projected climate-induced impacts for tourism in these regions[49].

Conversely, the water supply (E) and public administration and defence (O) sectors are generally adaptation-intensive, with ratios above 3.0 for most countries. Adaptation investment in water supply and sewerage is logical given its sensitivity to extreme weather events that induce irregularities in the access to water, and its links to flood protection infrastructure. Public sector investments are naturally expected to be quite high, since they include all adaptation spending carried out with public funds, as defined in the Methods. Moreover, mining and quarrying (B) is the most adaptation-intensive sector across our panel. Given the sector's relatively low GVA-in-GDP share across Europe, these results suggest high adaptation costs relative

to economic returns, pointing to the sector's vulnerability to climate-induced hazards[52]. Finally, agriculture, forestry and fishing (A) shows a clear divide among countries, with high ratios ( < 3.0) in relatively high-income states like Germany and UK, while lower-income states in Eastern and Southern Europe invest below their country's average. These lower adaptation intensities are remarkable given the high exposure of agricultural activities to climate-induced hazards in these countries[53]. Given the importance of subsidies to European agriculture[54] and that all agricultural adaptation measures—including those financed through subsidies—are part of the sector's CCA expenditures (see Supplementary Table 4), government policies can play a key role in mitigating these differences in adaptation intensity.

To elicit the speed of adaptation dynamics, we complement our descriptive analysis (Fig. 2) by computing the growth rates in adaptation expenditure for every sector-country combination ($n = 532$) during the 2018–2022 period. When accounting for inflation, we observe a relative deceleration in the year-on-year growth rate, which averages 37.4%—with standard deviation (SD) 5.0%—in 2019, and 30.6% (SD 6.4%) by 2022. However, when aggregating these yearly growth rates into the 19 economic sectors, we observe relatively little variation, with sectoral averages ranging between 28.4 and 32.6% in 2022. A summary table of the growth rates by sector and country can be found in Supplementary Note 3. Thus, while all sectors are considerably increasing their investments in adaptation every year, high-GVA sectors like Manufacturing and Wholesale and Retail Trade remain non-adaptation-intensive relative to the rest of the economy. As extreme weather events become more frequent and intense, increasing direct damages to high-value sectors could trigger larger, indirect macroeconomic effects[55] in vulnerable regions.

**Hazard-specific diversity of adaptation investments**

Finally, there is diversity in the type of hazards—and their frequency—that the 28 countries covered in our dataset may face. According to the Risk Data Hub maintained by the European Commission Disaster Management Knowledge Centre (EC DRMKC), there have been 1688 registered extreme

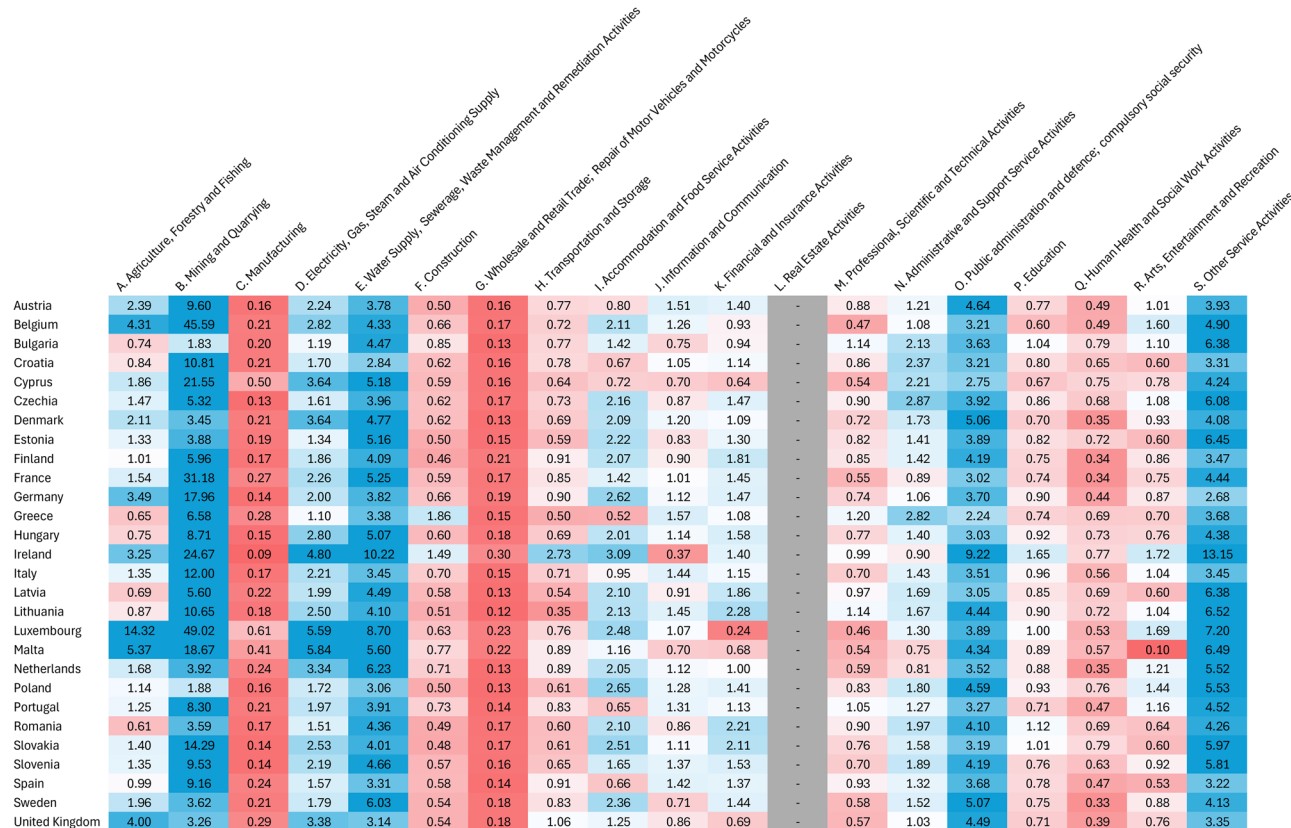

**Fig. 2 | Ratio of sectoral share in national adaptation investment to sectoral gross value added in GDP for the EU and the UK for 19 NACE 2 economic sectors.** The ratios are aggregated over all five hazard categories and averaged over the period 2018-2022. Note that the values presented below have been normalised by the size of each country's economy and its total investment in CCA, to aid inter-sectoral comparisons and discern trends across countries. If the ratio is equal to 1 (in white), the sector's expenditure in adaptation is exactly proportional to its contribution to the country's GDP. If the ratio is below 1 (in red), the sector is considered non-adaptation-intensive relative to the rest of the economy, whereas if it is higher than 1 (in blue), it is considered adaptation-intensive, as it invests more than its corresponding share in GDP. By construction, each row will always feature sectors with ratios above and below 1.0, as the ratios are defined per country, and thus their weighted average must equal 1.0. Note that the ratio for the sector L. Real Estate Activities has not been computed due to the inclusion of imputed rents in the estimation of GVA (a more extensive explanation is available in the Methods).

weather events since 2006[56]. We classify these events in five categories: Heatwaves (40 events), Droughts (3), Flooding (549), Wildfires (151) and Other (945). We clarify what is included in each category in the Methods. The economic losses arising from these events in the EU have been increasing in recent years[7], and thus an analysis of the adaptation action taken for each hazard type is warranted.

To address this gap, we estimate the relative share of adaptation expenditure per hazard type for the EU and the UK, averaged over the period 2018–2022 (Fig. 3). In all countries except Latvia and Luxembourg, sectors invest proportionately the most in adaptation to Heatwaves. In our dataset, heatwave adaptation measures include the installation of air-conditioning and ventilation systems, which are relatively affordable, undisruptive, and provide immediate direct benefits. The high average share of investments in adaptation to Heatwaves observed in the current snapshot of the data likely evens the picture of national adaptation investments, signalling relatively uniform shares (Fig. 1), and concealing geographical disparities accumulating in the long run. Yet, as expected, countries in Southern Europe dedicate some of the highest shares of adaptation expenditure to reduce the impacts of this hazard, as the need for cooling is dire. In general, countries that spend less on adapting to Heatwaves have a higher share in Flooding, with five countries dedicating over 40% of their adaptation investments to this hazard. Most landlocked states in Central and Eastern Europe fall into this category, with Austria, Slovakia, and Hungary all spending at least 39% on Flooding, and under 50% on Heatwaves. However, other riverine nations like Czechia and The Netherlands

do not follow this trend, suggesting an increased prioritisation of adaptation towards other hazards or sufficient past adaptation investments.

The sum of the remaining three categories does not reach 20% of adaptation expenditure for any of the 28 countries. In this group, Wildfires (1.6–3.9%) and Drought (0.5–1.2%) attract a relatively negligible share of investment. While wildfires have intensified in Europe in recent years, a considerable share of burnt areas sit within natural reserves[57], impacting ecosystems more than economic activity. Adaptation to drought is also relatively small because only the agricultural, manufacturing and public sectors are investing in related measures in our dataset. The Other category can attract a relatively higher share and shows greater variation across the countries in the dataset compared to other hazards, ranging from 5.3% in Sweden to 13.8% in Latvia. Within this umbrella category, windstorms can attract adaptation investment as they are often part of compound events with coastal flooding, while landslides coinciding with heavy rainfall are by far the most frequent hazard event in the EU and UK since 2006, with 582 separate incidents. Ultimately, the shares of all five categories remain relatively consistent, with a clear prioritisation of Heatwave and Flooding adaptation.

**The interplay between public and private adaptation**

So far, most CCA efforts have been in the domain of government-led public adaptation[58]. The level of public CCA might influence private investments in adaptation, either hindering or facilitating the latter. Understanding how public and private adaptation interact is essential to plan future investment

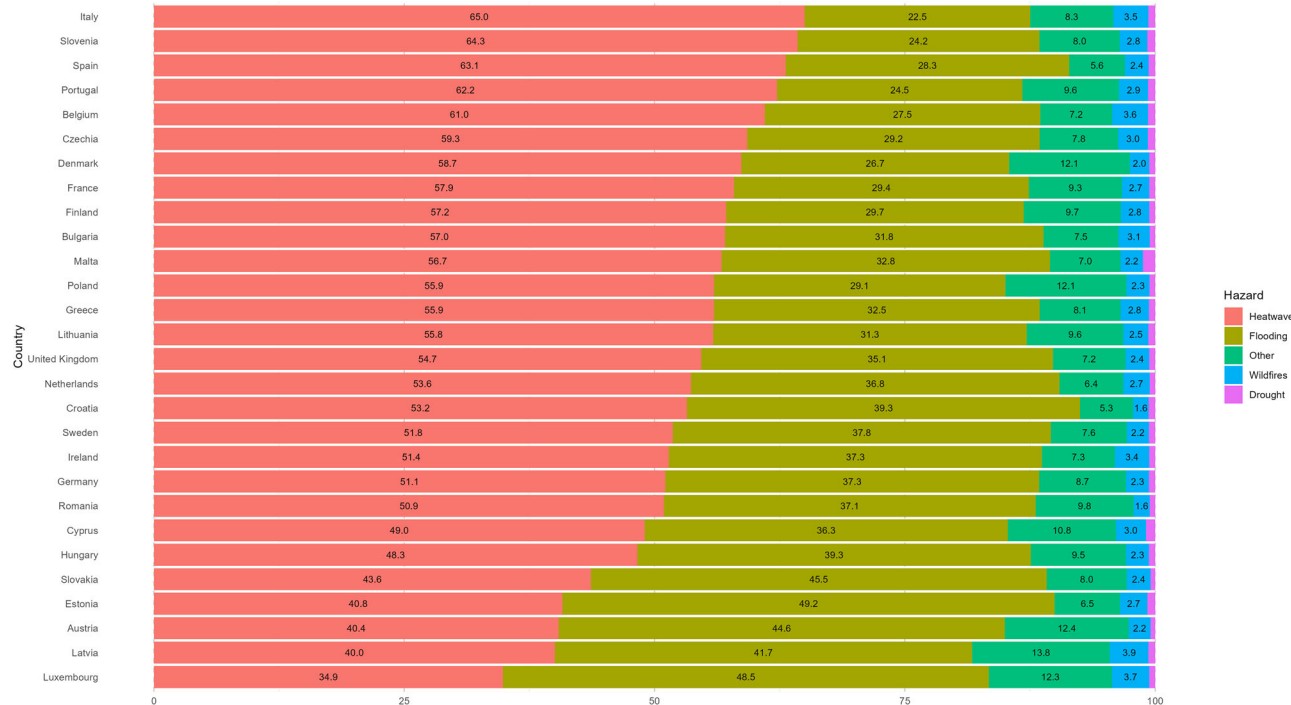

**Fig. 3 | Relative (%) share in total national adaptation expenditure for each of the five defined hazards (Heatwave, Flooding, Other, Wildfires, Drought).** The total national expenditure in adaptation is aggregated over the 19 economic sectors (NACE 2 Rev. A-S), and averaged over the period 2018–2022. The values presented inside the bar chart show the percentage share of total expenditure associated to each hazard in every country, helping to interpret which hazards attract most adaptation investment across the 28 countries. To ease readability, the countries are sorted top to bottom by relative expenditure on adaptation to heatwaves and the value of the Drought share is removed (it ranges between 0.5% for Slovakia and 1.2% for Malta).

for climate-resilient economies and to design public policies that effectively increase adaptation uptake across economic sectors. In our dataset, we have defined public adaptation as the spending undertaken by the sector Public administration and defence; compulsory social security (O) (Methods). Importantly, this includes adaptation investments directly paid by the government but not those indirectly funded through subsidies, as their public origin cannot be traced using the methodology developed by our data provider. In summary, public CCA investments represent the preparation and implementation of public adaptation measures, not entire national CCA plans.

The share of public adaptation in total adaptation is relatively constant across our panel, with a mean of 25.5% and an SD of 0.3%. For reference, the minimum share is 24.2% in Czechia in 2021, and the maximum share is 26.6% in Lithuania in 2018. Despite these similarities, we can leverage these data to establish whether there exists a relationship between public adaptation and the adaptation investments of individual private sectors in every country. There is an established literature that has conceptualised this relationship[17,42]. Initial evidence often suggested a crowding out effect of private investment in adaptation arising from moral hazard (i.e., low demand for private insurance if the government compensates flood damages[59]) or unintended consequences of public adaptation triggering private actions that lead to increasing risk[60]. At the same time, empirical evidence shows that beliefs that public government-led adaptations are inadequate discourage private household adaptation intentions[24], while PPPs can be leveraged to provide public-sector knowledge and research to build capacity for small businesses[41]. However, to the best of our knowledge, relationships between public CCA and private business adaptation investments have not been tested empirically.

To elicit this relationship, we use a fixed-effects panel regression model ($n = 8288$) relating the annual growth of private adaptation investment (dependent variable) to the growth of country-specific public adaptation (in nominal prices). Additionally, we control for a set of relevant country-specific socioeconomic and climatic variables, including GVA growth, the GDP deflator relative to 2018, the national average for population density, and the average sovereign debt rating. Additionally, we consider the number of events recorded for every hazard category in the previous 5 years in each country, to partially capture spikes in adaptation uptake following an extreme event[61]. The general specification and the source and interpretation of all control variables are found in the Methods.

Overall, we observe a positive relationship between the growth rates of public and private adaptation (0.279, standard error (se) 0.062, $p$-value = 0.000), such that an increase in public expenditure coincides with growth in private CCA investment. In percentage terms, we can interpret this result as follows: if public adaptation spending increases by 1%, private CCA investments are expected to rise by 0.28% over the same period. While this effect is relatively small, it suggests that public investments in CCA are analogous to other public investments, in that they do not discourage private expenditure (i.e., crowding in effect[62]). This result partially aligns with contemporary adaptation studies suggesting that greater trust in the government's response to a disaster positively influences private adaptation[24]. However, we cannot draw strong conclusions on whether it is a generalisable trend due to the relatively short temporal span of our panel (i.e., there are four available growth rates over a 5 year span).

Despite focusing on this public-private interaction, the remaining control variables provide further context as to what motivates growth in private adaptation investment (Fig. 4). The estimated coefficient corresponding to the level of private adaptation expenditure in the previous year is small and negative ($-0.0965$, se 0.0090, $p = 0.000$), implying that sectors that already invest highly in adaptation are modestly slowing down their growth in adaptation investments. Other socioeconomic variables present expected relationships with private adaptation growth. The coefficient for GVA growth is positive and significant (0.0303, se 0.0066, $p = 0.000$), implying that sectors whose GVA grows by 1%, increase their CCA investments by 0.03%. Moreover, sectors appear to update how much they

**Fig. 4 | Coefficient plot of the main regression (Eq. 5) on private adaptation investment growth.** The plot shows all independent variables, starting with public adaptation growth, while an insignificant constant term has been removed. The point estimate for every coefficient is represented by each dot, while the spikes show their 95% confidence intervals. All significant coefficients are in blue ($*p < 0.05$, $**p < 0.01$), while insignificant coefficients are shown in red. The public adaptation growth terms are defined as the first difference of the natural logarithms of public adaptation spending. The coefficients for private adaptation expenditure and the GDP deflator relate to the 1-year lag of these variables, as denoted by (t−1). The bottom four variables represent the number of hazard events of each category that occurred in a country in the previous 5 years.

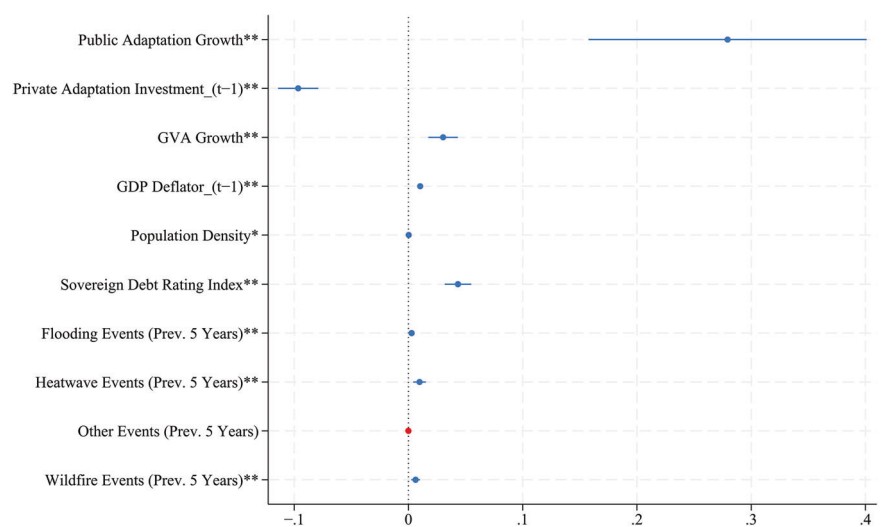

spend in CCA if they observe an increase in prices, as the coefficient associated to the 1-year lag of the GDP deflator—which is defined as the percentage increase in prices since 2018—is small and positive (0.0103, se 0.0010, $p = 0.000$). Additionally, population density is positively associated with the dependent variable (0.000241, se 0.000104, $p = 0.020$), pointing to the high need to protect large populations and valuable capital assets that cluster in climate-sensitive, urban zones[63]. Conversely, the relationship between private CCA growth and the sovereign debt rating index—a proxy for private investor trust in a country's financial position—is positive and significant (0.0434, se 0.0059, $p = 0.000$), suggesting that macro-financial stability can positively influence private CCA investment, at least in the EU and the UK.

Finally, we observe some variation arising from recent exposure to climate-induced hazards, which is defined at the country level. If we consider hazard events that have occurred in the previous 5 years, the number of events in the country shows a positive relationship with private CCA growth, specifically for Flooding (0.00279, se 0.00094, $p = 0.003$), Heatwaves (0.00969, se 0.00282, $p = 0.001$), and Wildfires (0.00628, se 0.00196, $p = 0.001$). Thus, private CCA investments grow in the aftermath of these events, either due to direct exposure and damage or as a precautionary action following occurrences in the same country. It is worth noting that while this effect for the Other category is insignificant (−0.0000196, se 0.0003387, $p = 0.954$), we could not conduct this assessment for Drought, due to the low number of observed drought events (3). Overall, our results show that private adaptation investments are sensitive to socioeconomic factors and recent exposure to climate-induced hazards, such that both channels should be considered when allocating public adaptation resources and designing adaptation uptake policies.

## Discussion

Increasingly frequent and extreme climate-induced hazards are leading to damage worldwide, including to the economies of the EU[7] and the UK. The IPCC highlights the need to increase climate action in the prevention of loss and damage[2], effectively calling for CCA to have a more prominent role in policy priorities of countries worldwide. In this context, our study provides an extensive overview of the current state of private-sector CCA investment in the EU and the UK, distinguishing between public and private adaptation, and presenting sector- and hazard-specific disaggregations for the 28 countries over 5 years (2018–2022). We find that adaptation expenditures are steadily increasing and outpacing GDP growth in Europe, led by The Netherlands and Southern European countries. Our sectoral analysis reveals consistent trends in adaptation intensity, including low intensity for manufacturing and high intensity for public services. Additionally, the most prioritised hazards by CCA expenditure are heatwaves and floods. Finally,

we observe a positive relationship between public and private adaptation growth, suggesting the presence of a small crowding in effect[62] of adaptation investments across our panel. These insights advance the quantitative understanding of empirical patterns in private-sector CCA adaptation investment along multiple dimensions (geographical, sectoral and hazard-specific) in Europe. As such, we address several key priorities for CCA research[20] and shed light on the potential drivers of future adaptation investments.

### A unique overview of CCA investment across Europe

Previous assessments of CCA initiatives across European countries have provided valuable insights at various geographical scales, especially at the municipal[45,64] and national levels[18,65]. However, these assessments typically lack sectoral detail or are limited to specific archetypes, such as coastal megacities[45]. Alternatively, other studies worldwide have focused on CCA enacted by specific sectors, such as water[66], manufacturing[29] and agriculture[30]. Yet, analyses of cross-sectoral adaptation are rare[67]. In the case of public adaptation, some countries directly disclose the planned budgets of major integrated adaptation projects, such as the Dutch National Delta Programme[68], although systematic analyses of total public adaptation spending are scarce[18,69]. Compared to these other macro-level studies, our analysis encompasses a wider range of hazard types (e.g., heatwaves), resulting in higher estimates shares of public spending in CCA (approximately doubling previously estimated shares for public CCA in Austria, Spain and The Netherlands[69]). In fact, our study goes beyond public CCA, shedding light on the overall state of CCA expenditure by including private investments. By analysing consistently sourced adaptation investment data across hazards, sectors, and countries, we discern common trends across 28 countries, allowing us to better understand how the private sector prioritises adaptation investments and how these interplay with public CCA in Europe.

### The need for sector-specific CCA strategies and incentives

We have found remarkable consistency in the intensity of adaptation investment by sector across the 28 countries. Across Europe, Manufacturing and Wholesale and Retail Trade, which when combined, represent around 30% of GDP in most countries[51] – have low adaptation intensity compared to the rest of the economy. Insufficient adaptation by these sectors can considerably harm their capital stock, especially in the event of a flood, affecting their ability to produce and trade high-value goods, reducing tax revenues for the government[18]. Moreover, tourism-adjacent sectors in Southern Europe also invest below the economy's average. As these service sectors are quite labour-intensive, climate-induced damages affecting tourism flows[49] can lead to spikes in unemployment and economic downturns. These differences in adaptation intensity call for the design of sector-

specific policies to foster autonomous CCA, considering the relative importance of each sector to the national economy, protecting their capital from physical damages, and their workforce from climate-induced productivity drops. Furthermore, as hazard events become more extreme and frequent, the direct damages experienced by exposed firms will only increase, triggering indirect impacts elsewhere in the economy[32,70], which can exceed the hazards' first-order effects[55]. Therefore, besides assessing sectoral exposure to climate-induced hazards, policymakers should consider sectoral adaptation intensities and the sector's criticality to the national economy when estimating benefit-cost ratios to allocate adaptation resources, as protecting these sectors will prevent the proliferation of climate damages through macroeconomic channels. In this endeavour, public CCA measures can also be leveraged to support these sectors.

## Synergies between public and private CCA

Early adaptation studies highlighted the risks of joint (public and private) adaptation, whereby structural, publicly funded CCA projects could discourage the uptake of more localised, private CCA measures[42], hindering efficient societal adaptation. While an assessment of CCA effectiveness is outside the scope of this study, our econometric analysis shows that public adaptation spending does not deter private investments in CCA. On the contrary, we find that public CCA spending mimics the trend of other public investments, especially those on infrastructure, and generally crowds in private investment across a variety of economic sectors and countries[62]. This aligns with empirical evidence that household adaptation intentions grow with public adaptation investments[24]. Our findings do not provide explicit recommendations on how to effectively direct public spending in CCA to enable private investments, as the mechanisms between adaptation measures are context-specific[38]. However, they can act as a starting point to broaden the uptake of public policies that already enable further private CCA[39–41]. This synergistic mechanism could be further exploited in combination with context-specific adaptation effectiveness assessments, which should help prevent misguided CCA investments and maladaptation[71]. Future work, including in-depth interviews with individual businesses revealing their motivation for CCA investments, could specifically explore which public adaptations are perceived by businesses as encouraging private action. Ultimately, both governments and firms are interested in the private sector efficiently achieving a sufficient level of adaptation (i.e., where capital assets are protected and labour productivity is not hindered), so that future investments can focus on growing businesses and creating jobs.

## Complementary socioeconomic assessments to deliver robust policy recommendations

Following the overview of current adaptation investments presented in this study, future work should prioritise socioeconomic studies that use more granular data to provide more targeted advice to decision-makers, including climate risk assessments for non-adaptation-intensive sectors. Firm-level studies would be particularly desirable to understand how firm size affects adaptation investments in different sectors, in line with previous work for manufacturing[29]. Additionally, other macro-level analyses of total adaptation expenditure would be desirable to validate our findings, while more systematic reporting of (economic) damages from extreme weather events would facilitate the attribution of spikes in adaptation uptake following an event. On a related point, studies that capture heterogeneity in the spatial distribution of capital could outline how sectors facing varying exposures to climate impacts[72] should adapt. In a nutshell, the key question of whether the current levels of adaptation intensity by sector are warranted should be answered swiftly. On the flipside, not all investments in CCA measures result in increased adaptive capacity and protection from future damages, including in the EU[73]. More empirical evidence on the effectiveness of different CCA measures is necessary to provide robust adaptation advice to both policymakers and businesses. Additionally, these assessments should explicitly consider the impact of these measures' timings (i.e., ex ante vs. ex post)[74], as proactive measures are generally considered to be more effective in protecting assets[75]. Determining whether the (lack of) adaptation

investment is justified by the physical exposure of the factors of production to climate impacts, and whether these CCA investments are effective for businesses, should be major points of emphasis for future research endeavours.

## Leveraging private CCA across macroeconomic contexts

Upscaling the results of this study to assess the macroeconomic effects of private CCA investments—e.g., using Computational Geneal Equilibrium modelling approaches[22,32]—could provide valuable insights to both researchers and policymakers. Additional synergies between public and private CCA could arise as more efficient national adaptation plans could reduce the burden of climate damages on the public budget[18], mitigating macro-financial risks including to sovereign debt[76]. Leveraging these synergies becomes even more important in the Global South, where the adaptation finance gap is widening for regions with limited financial resources[77] and greater exposure to climate impacts[46], and the scarcity of granular economic data remains a major barrier to develop more nuanced adaptation behavioural frameworks. In these contexts, understanding how public adaptation finance can support local businesses and enable autonomous adaptation can be essential. After all, an effective adaptation strategy involving all major macroeconomic stakeholders—households, firms and government—is essential to guarantee economic development and build a climate-resilient society.

## Methods
### Adaptation dataset

In this study, we have used a unique dataset that includes all CCA investments in the economies of 28 European countries, consisting of the EU and the United Kingdom. The coverage of this dataset has been chosen by considering the relative proximity and common governance frameworks and policies of these 28 countries (even after Brexit), in addition to the geographical and socioeconomic differences that lead to variance in exposure to different climate-induced hazards. To discuss broader regional trends across Europe, we follow the EuroVoc classification from the Publications Office of the EU, splitting the 28 countries as follows:

- **Northern Europe**: Denmark, Estonia, Finland, Latvia, Lithuania, Sweden.
- **Western Europe**: Austria, Belgium, France, Germany, Luxembourg, The Netherlands.
- **Central and Eastern Europe**: Bulgaria, Croatia, the Czech Republic, Hungary, Poland, Romania, Slovakia, and Slovenia.
- **Southern Europe**: Cyprus, Greece, Italy, Malta, Portugal, Spain.

The data is further split into the 19 economic sectors corresponding to Level 1 of the NACE Rev. 2 classification[78]. These sectors invest in adaptation towards five different hazard categories—Heatwaves, Flooding, Wildfires, Drought and Other—over the period between 2018 and 2022. The category Flooding includes investments in adaptation towards all three types of floods—coastal, pluvial and fluvial—while the category Other includes investments in adaptation to windstorms, landslides and sinkholes. A list of specific measures that fall into every hazard category are listed in Supplementary Note 2. Overall, spanning 28 countries, 19 economic sectors, five hazard categories and five periods, the dataset has a total of 13,300 observations; the complete list of hazards, sectors, countries and years can be found in Table 1.

The adaptation investment dataset was commissioned by the authors and developed by kMatrix Data Services, using a data consolidation approach that triangulates data from various transactional and operational sources to estimate economic activity[79], mitigating biases from each individual source. The dataset is constructed from the bottom-up, by implementing a taxonomy of adaptation measures related to a specific climate hazard type (see Supplementary Table 5), and aggregating over the relevant economic sector and country. To construct the dataset presented in this study, kMatrix drew from a compilation of over 27,000 independent datasets (both public and confidential) covering

**Table 1 | Summary of all four variable dimensions included in the adaptation expenditure dataset**

| Economic sector | Country | Hazard | Year |
|---|---|---|---|
| A. Agriculture, forestry and fishing | Austria | Drought** | 2018 |
| B. Mining and quarrying | Belgium | Heatwave | 2019 |
| C. Manufacturing | Bulgaria | Flooding | 2020 |
| D. Electricity, gas, steam and air conditioning supply | Croatia | Other | 2021 |
| E. Water supply; sewerage, waste management and remediation activities | Cyprus Czechia | Wildfire | 2022 |
| F. Construction | Denmark | | |
| G. Wholesale and retail trade; repair of motor vehicles and motorcycles | Estonia Finland | | |
| H. Transportation and storage | France | | |
| I. Accommodation and food service activities | Germany | | |
| J. Information and communication | Greece | | |
| K. Financial and insurance activities | Hungary | | |
| L. Real estate activities | Ireland | | |
| M. Professional, scientific and technical activities | Italy | | |
| N. Administrative and support service activities | Latvia | | |
| O. Public administration and defence; compulsory social security* | Lithuania Luxembourg | | |
| P. Education | Malta | | |
| Q. Human health and social work activities | Netherlands | | |
| R. Arts, entertainment and recreation | Poland | | |
| S. Other service activities | Portugal | | |
| | Romania | | |
| | Slovakia | | |
| | Slovenia | | |
| | Spain | | |
| | United Kingdom | | |

The economic sectors are all Level 1 NACE 2 economic sectors, while the countries span the EU and the United Kingdom. The total number of observations is 13,300. *Sector O is reclassified as a "public sector", and thus its observations are removed from the dependant variable in the regression. **The only two (private) sectors investing in Drought are A and C, and thus all other Drought observations are effectively set to 0.

most global financial transactions. It is worth noting that the estimates used in this paper draw from real firm transaction data and thus relate to realised expenditures, not planned investments. These estimates are produced by averaging at least seven different sources that have already been triaged for consistency (i.e., based on how accurate the estimates produced by each source have been in the past). An extensive description of the data consolidation process, as described by kMatrix Data Services, can be found in Supplementary Note 2. Among other applications, this data consolidation methodology has been used to analyse the Climate Services sector[44] in collaboration with a number of UK-based stakeholders (Greater London Authority, Imperial College London, UK Climate Change Committee), and as it relates to CCA, to assess the adaptation responses of global coastal megacities[45].

## Descriptive analysis

The three sets of results showing the state of adaptation investment in Europe have been produced by combining adaptation expenditure data obtained from kMatrix and macroeconomic aggregate data obtained from relevant national sources, including Eurostat[51] and the Office for National Statistics[80]. Firstly, total adaptation investment as a fraction of GDP for country $i$ in the year $t$ is defined in Eq. 1. For this, we sum over adaptation expenditures on all hazards $h$ and every sector $s$. For the sake of consistency throughout the analysis, the GDP in this section has been defined as the sum of the GVA of the 19 NACE 2 sectors. Note that the values for these metrics are equivalent in nominal and real terms (i.e., accounting for inflation) as they are ratios of nominal variables.

$$Adapt(\%GDP)_{i,t} = \frac{\sum_s \sum_h Adapt_{h,s,i,t}}{\sum_s GVA_{s,i,t}} \quad (1)$$

Secondly, to assess the relative position of each sector in terms of adaptation expenditure, we define an adaptation intensity ratio comparing sectoral shares in adaptation and in GDP. Our aim is to visualise any inter-sectoral trends arising from a sector's contribution to the national economy compared to its contribution to total adaptation investment. For this, we construct a ratio for a sector $s$ in a country $i$ between the sectoral share in national adaptation expenditure and the sectoral share in GDP, averaged over the period 2018–2022 (Eq. 2). In this metric, a sector that invests in adaptation proportionally to its size would have a ratio of 1, while sectors with ratios below 1 are considered to have low adaptation investment intensity, as explained for manufacturing in the Results (Fig. 2).

$$AdaptLevel_{s,i} = \frac{1}{t_{\max} - t_{\min}} \sum_t \frac{\sum_h Adapt_{h,s,i,t}/\sum_s \sum_h Adapt_{h,s,i,t}}{GVA_{i,s,t}/\sum_s GVA_{i,s,t}} \quad (2)$$

Note that the sector L. Real Estate Activities has not been included in this sectoral analysis (as seen in Fig. 2), as its adaptation investments cannot be compared to its GVA estimate due to the inclusion of imputed and private housing rents in national accounts' data. Imputed rents represent the equivalent sum of all rental income that house owners would pay to live in their own home, and are counted as GVA for this sector to account for differences in home ownership rates between countries. While imputed rents are available in Eurostat, private home rents (i.e., those paid from a tenant to a private home owner) are not, and thus rental income by households cannot be reliably deducted from total real estate GVA. Therefore, a reliable adaptation intensity ratio cannot be computed without private household adaptation data, which lie outside the scope of the analysis.

Finally, we examine hazard-specific adaptation investments across the countries in our panel. To do this, we define the relative share of every hazard type $h$ in total adaptation expenditure for every country, averaged over the period 2018–2022. In this exercise, we have aggregated adaptation investments over all sectors $s$ in a country $i$, such that we can compare country profiles, but not sector-specific profiles. The specific definition is presented in Eq. 3.

$$HazShare_{i,h} = \frac{1}{t_{\max} - t_{\min}} \sum_t \frac{\sum_s Adapt_{h,s,i,t}}{\sum_h \sum_s Adapt_{h,s,i,t}} \quad (3)$$

## Econometric analysis

The final component of the findings presented in this article focuses on the relationship between public and private adaptation investments in the EU and the UK. To achieve this, we leverage the coverage of the kMatrix adaptation expenditure dataset, which covers the aforementioned 28 countries, 19 Level 1 NACE Rev. 2 economic sectors, five different hazard types, and five calendar years, amounting to 13,300 observations. A full list of all possible dimensions in the dataset is presented in Table 1. However, to develop our regression analysis we focus on private adaptation investment

**Table 2 | Summary of all socioeconomic and climatic variables used to produce the descriptive and the econometric analyses**

| Variable name | Variable Type | Dimensions | Unit | Source |
|---|---|---|---|---|
| Private adaptation expenditure | Socioeconomic | Country, Sector, Hazard, Year | Million Euro | kMatrix Data Services |
| Public adaptation spending | Socioeconomic | Country, Hazard, Year | Million Euro | kMatrix Data Services |
| GVA | Socioeconomic | Country, Sector, Year | Million Euro | Eurostat, ONS |
| GDP deflator | Socioeconomic | Country, Year | Basis points (2018: = 100) | AMECO |
| Population density | Socioeconomic | Country, Year | Persons per Square Kilometre | Eurostat, ONS |
| Sovereign debt rating | Socioeconomic | Country, Year | Non-Dimensional | World Bank |
| Natural hazard events | Climatic | Country, Hazard, Year | Number of Events | Risk Data Hub (EC DRMKC) |

Besides the adaptation expenditure data, all other variables are obtained from publicly available sources. The category ''Dimensions'' lists the dimensions by which each data point can vary; these dimensions are shown in Table 1. For instance, Private Adaptation Expenditure data points are country-, sector-, hazard-, and year-specific, while Population Density only changes with country and year.

(our dependent variable), and thus isolate public investments by separating the O sector, using it as an explanatory variable instead. Note that other sectors can also be partially public, including healthcare (Q) and education (P) services. However, given that the public share of these sectors varies from country to country, we have treated them as private sectors in the analysis for simplicity. Additionally, given that only agriculture and manufacturing invest in private adaptation to drought, observations for all other sectors are effectively dropped for this hazard. After this filtering process, 8288 observations of private adaptation investment are considered.

In what follows, we seek to understand whether there is a crowding in (or crowding out) effect from public adaptation on private adaptation, whereby increases in public spending on CCA also result in an increase in private investments. To account for differences in the starting level of private adaptation investments, we include a 1-year lag of private adaptation expenditure. This is because we expect sectors with already high private CCA investment to slow down new investments, as businesses in that sector might have already implemented major adaptation measures. With that in mind, we build our core model in Eq. 4, using first differences to define private and public adaptation growth rates, in line with previous studies that assess how public and private investments interact[62].

$$\Delta\left(\ln PrivAdapt_{h,s,i,t}\right) = \beta_1\Delta\left(\ln PubAdapt_{h,s,i,t}\right) + \beta_2\ln PrivAdapt_{h,s,i,t-1} + u_{h,s,i,t}$$

(4)

In the basic formulation, the estimated coefficient $\beta_1$ relates the growth rate of public adaptation spending to its private adaptation counterpart; crowding in occurs if $\beta_1$ is positive, while crowding out occurs if $\beta_1$ is negative. The estimated coefficient $\beta_2$ shows whether pre-existing adaptation investments affect new investments; if $\beta_2$ is negative, then the growth of CCA investment slows down as the sector adapts over time.

We continue to develop the core model presented to capture other relevant dynamics present in this dimensional space. We refine our model by adding a set of socioeconomic indicators as control variables that could embed the overall economic performance of the sectors and countries in our panel, as well as capture some of the spatial characteristics that are relevant to the economics of CCA. Firstly, we introduce GVA growth (calculated using first differences analogously to the adaptation variables) to capture sector- and country-specific variation, as we expect a growing sector to be more likely to increase investments generally. For the real estate sector, we assume that overall GVA growth is a representative estimate of the growth in GVA of real estate companies, including housing corporations. Secondly, given the presence of inflationary pressure on the European economy by 2022, we include the 1-year lag of the GDP deflator (based on 2018 prices), in case businesses substantially change their investments following sharp price level changes; these values are extracted from the AMECO database maintained by the European Commission[81]. To provide further context related to the spatial distribution of the economies considered, we add the yearly national average of population density to the model[82,83]. As discussed in the economic agglomeration literature, geospatial characteristics can

influence the exposure of modern societies to climate hazards[63]; this is particularly relevant in large climate-sensitive urban zones. Given that our adaptation data is national, we use the national average of population density as a proxy for agglomeration. Additionally, we incorporate the macro-financial dimension by adding the sovereign debt rating of each country as defined by the World Bank, expressed as a numerical index of the ratings given to each country by the Big 3 credit rating agencies: Moody, Fitch, and Standard&Poor[84].

Finally, as we expect recent climate-induced hazard events to trigger a strong increase in adaptation expenditure, we include as a final explanatory variable the number of hazards events in the previous five years in the country—as available in the EC DRMKC Risk Data Hub[56]. To explore the desired effect, this indicator was interacted with a dummy variable for every specific hazard.

Adding all of these contextual variables together, and constructing fixed effect clusters for each hazard, sector and country combinations, the complete econometric model used in the analysis is presented in Eq. 5. Some key characteristics of the variables used in this model, including their sources[51,56,80–84], are summarised in Table 2.

$$\begin{aligned}\Delta\left(\ln PrivAdapt_{h,s,i,t}\right) = {}&\alpha_0 + \beta_1\Delta\left(\ln PubAdapt_{h,s,i,t}\right)\\ &+ \beta_2\ln PrivAdapt_{h,s,i,t-1} + \beta_3\Delta\left(\ln GVA_{s,i,t}\right)\\ &+ \beta_4 GDPdef_{i,t-1} + \beta_5 PopDens_{i,t} + \beta_6 SovDebt_{i,t}\\ &+ \sum_h \beta_{7,h}EventCount5yr_{h,i,t} * hazard_h + FE_{h,s,i} + u_{h,s,i,t}\end{aligned}$$

(5)

## Sensitivity analysis

To assess the sensitivity of our model, we conduct a jackknife analysis. In this method, we systematically exclude one unit at a time—such as a country, sector, hazard, or year—and assess the stability and robustness of our results. This technique helps us identify whether any single unit has a disproportionate influence on the model's outcomes. Our analysis indicates that excluding any specific country, sector, hazard (except drought), or year does not drastically change our estimation results. When we estimate our regression effects by individual hazard type, we observe that public CCA investment growth has a negative effect on private CCA investment growth for all hazards except drought (see Supplementary Table 9), contrasting with the results of our primary regression. However, upon further analysis, this discrepancy appears to be time-dependent. To explore this, we analysed the interaction between public and private CCA investments by year and hazard (as presented in the Time-Hazard column in Supplementary Table 9). This analysis reveals that after 2021 the effect of public investments on private investments becomes markedly positive across all hazards. This shift helps explain the overall positive average effect observed in our main analysis when drought is included, as the positive effect of drought remains stable across years, while the effects of other hazards transition from negative to positive over time. This time-dependence provides a possible explanation for the nuanced relationship between public and private CCA investments in the context of evolving hazard impacts.

Furthermore, while our regression associates public and private investments, reverse causality—where the effect can work both ways—might be a concern. However, in the case of CCA investments, this issue is mitigated. Historically, CCA initiatives have been led by governments (national and regional) with minimal investments from the private sector[58]. Government CCA investments are planned well in advance and are publicly available through budget plans. In contrast, private investments in CCA are not publicly known, and only a fraction of firms report them. Therefore, it is unlikely that the public sector adjusts its CCA investments each year in response to changes in private CCA investments during the same period.

## Data availability
The public and private adaptation expenditure data that support the findings of this study are available from kMatrix Data Services but restrictions apply to the availability of these data, which were used under license for the current study, and so are not publicly available. Any queries regarding the raw data should be directed to kMatrix Data Services (enquiries@kmatrix.org). All other data that support the findings of this study, including those used to generate the presented figures, are available in the Zenodo repository: https://doi.org/10.5281/zenodo.14288580.

## Code availability
The R code used to produce the visualisations and the STATA code used for the regression are available in the same Zenodo repository: https://doi.org/10.5281/zenodo.14288580.

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

## Acknowledgements
This work was supported by the European Research Council under the EU's Horizon 2020 Research and Innovation Programme (Grant Agreement no. 758014). The authors would like to thank kMatrix Data Services, specifically Sarah Howard and Steve Howard, for their continuous support with the data administration.

## Author contributions
I.C.A. conducted the descriptive analysis, produced the data visualisations, co-developed the regression model, and drafted the manuscript. T.C. co-developed the regression model, conducted the sensitivity analysis, and co-wrote the manuscript. S.S. reviewed the regression analysis, helped to contextualise the results, and edited the manuscript. O.I. supported the descriptive analysis, provided policy-relevant context for the discussion, and edited the manuscript. T.F. designed the overall research project, conceived the research design, outlined the introduction, and co-wrote the manuscript.

## Competing interests
The authors declare no competing interests.
