## [Peer review file · Communications Earth & Environment]

Private investments in climate change adaptation are increasing in Europe, although sectoral differences remain

Corresponding Author: Mr Ignasi Cortés Arbués

Version 0:

Decision Letter:

Dear Mr Cortés Arbués,

Your manuscript titled "Patterns in climate change adaptation spending in the European private sector" has now been seen by 2 reviewers, and we include their comments at the end of this message. They find your work of interest, but some important points are raised. We are interested in the possibility of publishing your study in Communications Earth & Environment, but would like to consider your responses to these concerns and assess a revised manuscript before we make a final decision on publication.

We therefore invite you to revise and resubmit your manuscript, along with a point-by-point response that takes into account the points raised. Please highlight all changes in the manuscript text file.

Please submit your point-by-point responses as a separate file, distinct from your cover letter where you can add responses to the Editors' comments that you do not want to be made available to the reviewers. Word files are preferred. We recommend that any figures, tables or graphs that are included in the response to reviewers are also included in the main article or Supplementary Information.

Please use the following link to submit your revised manuscript, point-by-point response to the referees' comments (which should be in a separate document to any cover letter), a tracked-changes version of the manuscript (as a PDF file) and the completed checklist:

Link Redacted

We hope to receive your revised paper within six weeks; please let us know if you aren't able to submit it within this time so that we can discuss how best to proceed. If we don't hear from you, and the revision process takes significantly longer, we may close your file. In this event, we will still be happy to reconsider your paper at a later date, as long as nothing similar has been accepted for publication at Communications Earth & Environment or published elsewhere in the meantime.

Please do not hesitate to contact us if you have any questions or would like to discuss these revisions further. We look forward to seeing the revised manuscript and thank you for the opportunity to review your work.

Best regards,

Alienor Lavergne, PhD
Senior Editor
ORCID: 0000-0002-4591-1217
Communications Earth & Environment

EDITORIAL POLICIES AND FORMATTING

Editorial Policy: [Policy requirements](https://www.nature.com/documents/nr-editorial-policy-checklist.pdf) (Download the link to your computer as a PDF.)

- Behavioural and social science
- Ecological, evolutionary & environmental sciences
- Life sciences

<https://www.nature.com/documents/nr-reporting-summary.zip>

Furthermore, please align your manuscript with our format requirements, which are summarized on the following checklist: [Communications Earth & Environment formatting checklist](https://www.nature.com/documents/commsj-phys-style-formatting-checklist-article.pdf)

and also in our style and formatting guide [Communications Earth & Environment formatting guide](https://www.nature.com/documents/commsj-phys-style-formatting-guide-accept.pdf) .

*** DATA: Communications Earth & Environment endorses the principles of the Enabling FAIR data project (<http://www.copdess.org/enabling-fair-data-project/>). We ask authors to make the data that support their conclusions available in permanent, publically accessible data repositories. (Please contact the editor if you are unable to make your data available).

All Communications Earth & Environment manuscripts must include a section titled "Data Availability" at the end of the Methods section or main text (if no Methods). More information on this policy, is available at <http://www.nature.com/authors/policies/data/data-availability-statements-data-citations.pdf>.

If a community resource is unavailable, data can be submitted to generalist repositories such as [figshare](https://figshare.com/) or [Dryad Digital Repository](http://datadryad.org/). Please provide a unique identifier for the data (for example a DOI or a permanent URL) in the data availability statement, if possible. If the repository does not provide identifiers, we encourage authors to supply the search terms that will return the data. For data that have been obtained from publically available sources, please provide a URL and the specific data product name in the data availability statement. Data with a DOI should be further cited in the methods reference section.

REVIEWER COMMENTS:

Reviewer #1 (Remarks to the Author):

Dear authors,

Thank you for giving me the opportunity to review this very interesting manuscript. The topic of the manuscript fits well with the objectives and scope of "Communications Earth & Environment". Overall, the paper addresses a very important research gap, is well elaborated, and provides very relevant results. The statistical approach is also well and transparently presented. Overall, the paper is concise, well written and structured, making a clear and important contribution to the literature. I have only some minor recommendations to improve the paper:

1. I suggest replacing the partly contested term "natural hazard" with the term "extreme weather events".

2. It would be good if the authors provide a definition of “adaptation economy”.
3. The findings that “a relatively even level of 120 adaptation spending across the EU and the UK” (line 121) is surprising, as it can be assumed that the spending is spatially more diverse, since the countries are affected differently. Perhaps the authors can further discuss their finding.
4. Given the crowding in effect, public spending may correspond to pressure on the private sector for further investment in CCA, so that the private sector may not be acting voluntarily.
5. Since some European regions are grouped together as southern or eastern Europe in the text, it might be helpful for the reader if an additional column were added to Table 2 to categorize the countries into these geographic categories.

Reviewer #2 (Remarks to the Author):

The manuscript offers insights into the current situation of adaptation expenditure in the private sector in European countries. It gives details on sectoral investment shares and aims at finding sectors with “underinvestment” to guide adaptation policy. The analysis mostly gives descriptive statistics, but also some correlations trying to answer the complex question of what drives investments in private adaptation. A key finding is that public adaptation crowds-in private adaptation. The topic is very relevant as information on climate change adaptation in the private sector is scarce and mostly based on case studies. Yet, despite its high relevance, I cannot recommend publication of the manuscript in its current state. I have several major concerns and also some minor points which need to be addressed, before it can be considered for publication.

Major issues:

1. The database (“kMatrix”) is not described transparently enough. In the SI the authors give a reference to a scientific paper, but this is not sufficient. My concern is that some of the results could simply be the result from the method that was applied to construct the database itself.
2. Very often the manuscript is speculative and offers “results” that are not based on empirical findings. I will list some examples below. I urge the authors to reflect on this.
3. The descriptives are interesting, but things are sometimes not interpreted correctly (or overinterpreted).

General comments:

1. The language is sometimes too emotional (e.g. “has never been more important” in the abstract, or “unfortunately” in line 47; “greater trouble” in line 174 etc.). The authors should use a more scientific language.
2. From my perspective the key finding of the paper is the crowding-in of private investment and this could be strengthened throughout the whole paper.

Detailed comments by section:

1. Title: Usually the term “spending” is used in the context of public expenditure, so I suggest using “adaptation investment” or the like instead.
2. Abstract: What’s the difference between welfare and well-being? More clarity is needed in the abstract. Also, it is not clear what the “adaptation economy” is. I suggest to not use this term.

Intro:

3. Additional to source 3) more references are needed. At least the IPCC should be cited, but ideally a bunch of original research papers.
4. second para is lengthy. I suggest splitting it up.
5. Line 77: public sector not only “delivers” (I think what the authors mean is “supply”) infrastructure but also protects it.
6. Lines 81-85: the social/equity dimension of CCA is important to raise here. The question is also whom the government protects and how to use scarce resources from that perspective.
7. Line 99ff: are foreign direct investments included in the sectoral investments?
8. The wording of the “adaptation economy” that is “growing steadily” sounds very positive (à la increases GDP), but I think this is not the message that the authors want to deliver. It is important to stress that adaptation investment is a kind of cost to the macroeconomy, since the resource could be used elsewhere more productively. Very often the only return on investment in adaptation is the avoided or reduced damages.

Results:

9. line 113: it is 243% not 143%
10. line 116: “level” should be “share”
11. The first paragraph ends with a lot of speculation. A more profound reference basis is needed for many of the claims (e.g. ref 43 is certainly not enough, also because it is very outdated).
12. lines 128ff: Having a higher adaptation level in place is eventually the same as being relatively less impacted.
13. Much of the results could also be explained by income differences. A low share of GDP of a rich country might be sufficient for the country’s adaptation requirements, whereas a high share of GDP of a lower income country might not. This dimension is currently underdeveloped.
14. Fig. 1. Isn’t Denmark also heavily affected by sea level rise, so this same as for the Netherlands.
15. lines 150ff: I strongly recommend to not use the terms “underinvestment” and “overinvestments” in this context. In fact, what the analysis shows is simply a ratio, but this does not say anything about the requirement of a sector’s adaptation. For some sectors a very low share might be perfectly fine, whereas for some others an already high share might still be insufficient. What would be needed is a spatial mapping of sectoral activity and an overlay with hazards to then derive the

risk and adaptation requirements by sector. In fact, it can never happen – by construction – that all cells in Table 1 turn green.

16. The “Real estate” sector needs more care. What does this sector really show? Is it only the service provision, or does it really include all the assets (buildings)?

17. Sector “Water supply”. Note that this sector is closely connected to flood protection infrastructure.

18. Line 173 “may signal” – this is speculative

19. Line 207 “per hazard” is confusing. It sounds like the authors divided by the number of events, which is not the case, I assume (rather it is the attribution of an adaptation investment to a specific hazard type).

20. Lines 209ff: I would not use the term “attract”, since no causality is proven. It simply gives an approximation of the main risks at the moment. The key question to answer would be whether an increase in risks really leads to increased investment in adaptation, but this would require a longer time series. But maybe the dataset allows to look into extreme events and check whether investments have increased in the subsequent years (for specific country-sector-hazard combinations).

21. Lines 249ff: This requires some cross-checks to other studies.

22. On the interplay between public-private: This is very interesting, but deserves at least more discussion. What about public-private-partnerships, for example?

23. Lines 289ff: The slow-down in investments might be due to the fact that adaptation happens in waves; and often only after an extreme event. This needs at least discussion.

24. Line 296: How is population density included? National average?

25. Line 313: “should be considered” – say how!

Discussion:

26. Line 360: “underinvestment” is not measured properly, so an increase based on this index might be inefficient (see my comment on that above). C-B ratios would be needed, ideally including economy-wide effects.

27. I expect a thorough update of the discussion section after the revisions.

Methods

28. As given as a major point of concern above, a detailed description of how the dataset was constructed is needed.

Communications Earth & Environment is committed to improving transparency in authorship. As part of our efforts in this direction, we are now requesting that all authors identified as ‘corresponding author’ create and link their Open Researcher and Contributor Identifier (ORCID) with their account on the Manuscript Tracking System prior to acceptance. ORCID helps the scientific community achieve unambiguous attribution of all scholarly contributions. You can create and link your ORCID from the home page of the Manuscript Tracking System by clicking on ‘Modify my Springer Nature account’ and following the instructions in the link below. Please also inform all co-authors that they can add their ORCIDs to their accounts and that they must do so prior to acceptance.

Version 1:

Decision Letter:

<*** REMEMBER TO ATTACH REVISIONS CHECKLIST (WORD)***>

Dear Mr Cortés Arbués,

Your manuscript titled "Patterns in climate change adaptation investment in the European private sector" has now been seen by our reviewers, whose comments appear below. In light of their advice we are delighted to say that we are happy, in principle, to publish a suitably revised version in Communications Earth & Environment.

We therefore invite you to revise your paper one last time to address the remaining concerns of our reviewers. At the same time we ask that you edit your manuscript to comply with our format requirements and to maximise the accessibility and therefore the impact of your work.

EDITORIAL REQUESTS:

****Please take care to match our formatting and policy requirements. We will check revised manuscript and return manuscripts that do not comply. Such requests will lead to delays. ****

SUBMISSION INFORMATION:

OPEN ACCESS:

Communications Earth & Environment is a fully open access journal. Articles are made freely accessible on publication. For further information about article processing charges, open access funding, and advice and support from Nature Research, please visit <https://www.nature.com/commsenv/open-access>

Link Redacted

Best regards,

Alienor Lavergne, PhD
Senior Editor, Communications Earth & Environment
Consulting Editor, Communications Sustainability
ORCID: 0000-0002-4591-1217

Springer Nature
The Campus, 4 Crinan Street, London N1 9XW, UK
www.nature.com/commsenv
[@commsearth.bsky.social](https://www.instagram.com/commsearth.bsky.social)

REVIEWERS' COMMENTS:

Reviewer #1 (Remarks to the Author):

The authors have carefully considered the reviewers' comments and presented the changes transparently. I am satisfied with the changes, which have significantly improved the manuscript, particularly the discussion of the results. Therefore, I have no further comments and recommend publication.

Reviewer #2 (Remarks to the Author):

The authors have taken up all of my concerns and have addressed them satisfactorily.

I only have some very minor points, which I'd like to see resolved, before publishing.

In Table 3 is SI, some entries in the "DATA sources" column only are labelled with a number. Why? What does that mean? Are they confidential?

The sentence: "protect both the population and the economy" – I think this is too narrow. What about ecological systems, biodiversity etc.? Please rephrase.

At some points the term "firms" is in fact wrongly used, as the analysis focuses on economic sectors. Of course, "firms" can be used in the arguments etc., but not when e.g. saying what you present. For example in the discussion: "our study provides an extensive overview of the current state of private CCA investment by firms...", which is not true.

Concerning the low intensities in sector A: could it be that the public sector steps in here a lot and protects this sector (via flood protection for example)? What about subsidies for adaptation (e.g. for irrigation infrastructure)? Is this included? (Sorry for not having brought this up in the initial review). Maybe a sentence or two can be added to the manuscript?

Response to Reviewers – Patterns in climate change adaptation investment in the European private sector

Please note that the line numbers cited in this document match those of the revised manuscript in track changes, not the clean document.

Reviewer #1 (Remarks to the Author):

Dear authors,

Thank you for giving me the opportunity to review this very interesting manuscript. The topic of the manuscript fits well with the objectives and scope of “Communications Earth & Environment”. Overall, the paper addresses a very important research gap, is well elaborated, and provides very relevant results. The statistical approach is also well and transparently presented. Overall, the paper is concise, well written and structured, making a clear and important contribution to the literature. I have only some minor recommendations to improve the paper:

Response: Thank you very much for your kind words and appreciation for this topic. We are also thankful for your recommendations regarding terminology. We agree with most of them and have implemented your feedback throughout the manuscript. We hope that the revised manuscript resolves these questions.

1. I suggest replacing the partly contested term “natural hazard” with the term “extreme weather events”.

Response: Thank you for pointing out this important clarification on terminology. We have removed all reference to “natural hazards” in the manuscript and have used “extreme weather events” or “climate-induced hazards” where relevant.

2. It would be good if the authors provide a definition of “adaptation economy”.

Response: We thank Reviewer #1 for highlighting the need to delineate the ‘adaptation economy’. In the original manuscript, we meant it as a network of transactions and monetary flows that relate to the provision of climate adaptation goods and services (i.e., who or which sectors buy adaptation AND from whom). The revised manuscript focuses more strongly on which sectors are committing the expenditure. Considering this, and the strong recommendation from Reviewer #2 to avoid the term altogether, the revised manuscript now does not use this term anymore. Instead, we provide more concise terminology where appropriate. We hope the text is now more clear for a general reader of *Nature Communications Earth & Environment*.

3. The findings that “a relatively even level of 120 adaptation spending across the EU and the UK” (line 121) is surprising, as it can be assumed that the spending is spatially more diverse, since the countries are affected differently. Perhaps the authors can further discuss their finding.

Response: Thank you for the opportunity to clarify this. We agree that this result seems surprising given the geographical and climatic diversity within Europe. However, this relatively even level of national adaptation as a share of GDP could be explained by the relative importance of adaptation to heatwaves (compared to other hazards) in most countries in the dataset (i.e., in Figure 2, 21 out of 28 dedicate more than 50% of investments to heatwaves). While we do not know the relative expenditure share of individual CCA measures within a hazard type, we expect that most adaptation to heatwaves happens in the form of ventilation and air conditioning, which are relatively affordable, undistruptive, and provide direct benefits. The revised manuscript now clarifies this in the section "Hazard-specific diversity of adaptation investments" when discussing Figure 2. Specifically, in **Lines 264-269** we state that:

"In our dataset, heatwave adaptation measures include the installation of air-conditioning and ventilation systems, which are relatively affordable, undistruptive, and provide immediate direct benefits. The high average share of investments in adaptation to *Heatwaves* observed in the current snapshot of the data likely evens the picture of national adaptation investments, signalling relatively uniform shares (**Figure 1**), and concealing geographical disparities accumulating in the long run."

4. Given the crowding in effect, public spending may correspond to pressure on the private sector for further investment in CCA, so that the private sector may not be acting voluntarily

Response: Thank you for the comment. We agree that this might be an interesting phenomenon to explore, especially via more detailed work, including in-depth interviews with individual firms to better understand their motivation for CCA investments. While we do plan such analysis for future work, here we have reached the limit in our analysis of what this dataset can tell about the relationship between private and public adaptation. Nevertheless, the EU does not have a compulsory reporting mechanism similar to that for GHG emissions, and we are not aware of regulatory or market-based instruments that governments in Europe currently use to impose pressure on businesses to invest in private adaptation. European governments can only provide information in the hope of nudging the private sector to invest in their own climate resilience. Therefore, the investments observed can overall be considered to occur autonomously, driven by one's own private interests, implying that the suggested interaction is unlikely, at least for now.

The revised manuscript now adds a brief note about the need to address this in future work (via interviews with individual businesses) on **Line 464-467**. Furthermore, given that 'autonomous adaptation' is the recurring term used in the literature to describe CCA initiatives undertaken by households and businesses (**Lines 89-92**), we adopt it in our manuscript and have added it where relevant (i.e., when discussing the overall findings at the end of the introduction in **Line 124**).

5. Since some European regions are grouped together as southern or eastern Europe in the text, it might be helpful for the reader if an additional column were added to Table 2 to categorize the countries into these geographic categories.

Response: Thank you for the suggestion, including where to clarify it in the text. To address your comment, we have now adopted the EuroVoc classification from the Publications Office of the European Union and have listed the categories and countries in the Methods (**Lines 525-532**). When mentioned, these groupings are now highlighted in italics (**Lines 147, 155**). We

have not edited Table 2 as it specifically addresses the dimensions covered in the econometric analysis.

Reviewer #2 (Remarks to the Author):

The manuscript offers insights into the current situation of adaptation expenditure in the private sector in European countries. It gives details on sectoral investment shares and aims at finding sectors with “underinvestment” to guide adaptation policy. The analysis mostly gives descriptive statistics, but also some correlations trying to answer the complex question of what drives investments in private adaptation. A key finding is that public adaptation crowds-in private adaptation. The topic is very relevant as information on climate change adaptation in the private sector is scarce and mostly based on case studies. Yet, despite its high relevance, I cannot recommend publication of the manuscript in its current state. I have several major concerns and also some minor points which need to be addressed, before it can be considered for publication.

Response: Thank you very much for your time and extensive comments that have helped us to substantially improve the paper. Particularly, we appreciate the specificity of the comments and their high level of detail. Moreover, we thank Reviewer #2 for highlighting the importance of the topic, especially in the context of data scarcity for private CCA. We have taken all the suggestions seriously and addressed them carefully as we explain point by point below.

Major issues:

1. The database (“kMatrix”) is not described transparently enough. In the SI the authors give a reference to a scientific paper, but this is not sufficient. My concern is that some of the results could simply be the result from the method that was applied to construct the database itself.

Response: Thank you for the opportunity to clarify this methodological point in more detail. Given that the macro-level data on private CCA is practically non-existent, we agree with Reviewer #2 on the importance of data transparency. Therefore, we have taken your concern seriously and addressed it in the revised manuscript through a number of steps. Firstly, in **Lines 59-60** of the revised manuscript, we acknowledge that:

“While there are no perfect datasets, their unavailability in the domain of private CCA is exceptional.”

Secondly, the revised manuscript now provides a fully worked example of a data point in our database (which can be found in the **Supplementary Information, p.3-21**), to increase the transparency of the methodology amongst both reviewers and readers. This detailed example, which relates to the Agricultural sector in The Netherlands in 2018, presents 42 distinct sources, some of which are public and some of which are confidential, which are combined to produce one single data point (this process is repeated to complete the entire 28-country, 19-sector, 5-hazard type dataset). Secondly, this specific example, kMatrix has provided an extensive description of their data triangulation methodology, which includes the six stages of their multi-sourcing approach. We have added this explanation with limited editing to the Supplementary Information. Finally, we have clarified some of the key components of the kMatrix methodology in the Methods, explicitly mentioning the bottom-up approach used to construct each data point at the country, sector, hazard level (**Lines 542-554**). We hope that these detailed methodological clarifications help the reader better grasp the nature of the data.

2. Very often the manuscript is speculative and offers “results” that are not based on empirical findings. I will list some examples below. I urge the authors to reflect on this

Response: We apologise if the previous version of the text gave this impression, as this was certainly not our intention. We have now carefully proofread the manuscript to ensure that there are no statements that are unsupported by our empirical findings and have actively removed more speculative wording (see for example our response to Detailed Comment #11). Additionally, among other changes, we have carefully revised the interpretation of sectoral trends by referring to adaptation intensity instead of “over-/under-investment”, as this cannot be observed empirically without including exposure to extreme weather events by the affected sectors as part of the analysis (in relation with your Comment #15). We have addressed all other relevant examples provided by Reviewer #2 in our responses to the detailed comments below. Finally, we have more explicitly delineated the scope of our findings in the Discussion, stressing the need of future work to combine our sectoral results with a granular analysis of exposure to climate-induced hazards, and more nuanced implications on the potential of the crowding-in mechanisms.

3. The descriptives are interesting, but things are sometimes not interpreted correctly (or overinterpreted).

Response: We appreciate the opportunity to address some of these misunderstandings and for the specific points raised in the “Detailed Comments” section. We thank you for your interest in the descriptive analysis. Currently, the field is missing even a simple descriptive analysis of what businesses do regarding climate adaptation, and how it differs per sector, country, or hazard. As we highlight in **Lines 67-69**:

“our understanding of how much businesses invest in CCA and what these patterns look like across economic sectors and countries remains limited”

Therefore, we believe that descriptive statistics are one of the most important novel contributions of this manuscript. They offer a unique overview on comparable, cross-country data on adaptation expenditures, especially at the sector level, not publicly available to scholars/practitioners to date.

Moreover, We have addressed the Detailed Comments that relate to overinterpretation below, placing specific emphasis on #8 (on the positive implications of adaptation spending growth), #11 (on future implications of current adaptation expenditure levels), #15 (on the use of under-/over-investment for sectors), and #20 (on inferring causality in the hazards analysis). We refer the Reviewer to our specific mitigation strategy on each comment separately.

General comments:

1. The language is sometimes too emotional (e.g. “has never been more important” in the abstract, or “unfortunately” in line 47; “greater trouble” in line 174 etc.). The authors should use a more scientific language.

Response: Thank you for pointing out this necessary change in language register. We have addressed the examples mentioned above and made the language more neutral (for example: “has never been more important”  “there is an increasing need for” (Lines 14-16); “Unfortunately”  “Still” (Line 51); “may signal greater trouble”  “pointing to a sector’s vulnerability” (Lines 210-212)). Moreover, we have made additional changes throughout the text to reflect a more scientific tone.

2. From my perspective the key finding of the paper is the crowding-in of private investment and this could be strengthened throughout the whole paper.

Response: We thank Reviewer #2 for stressing the importance of the crowding-in mechanism, given its potential to speed up adaptation uptake in the private sector. In general, we believe that our contribution to the literature with this piece is two-fold. Firstly, as explained in our response to your Major Issue #3 above, the descriptive statistics themselves are an important advance due to the lack of systematic data on CCA investments, especially at the macro-level. Specifically, our findings could aid economic analyses as a starting point to calibrate adaptation expenditure in simulation models, including input-output or computational general equilibrium models used by policymakers in ministries (this point is stressed in Lines 506-509). Additionally, we highlight the importance of providing this unique overview in the Discussion section, including through a “wider range of hazard types” (Line 419) by going “beyond public CCA, shedding light on the overall state of CCA expenditure by including private investments” (Lines 421-423).

Secondly, we agree that the crowding-in mechanism could be strengthened and have given it a more prominent role in the Introduction and Discussion. In the Introduction, we now refer to a few examples in which public CCA has shown potential to signal the need for private CCA and has increased the adaptive capacity of small businesses, including through public-private partnerships (Lines 90-93). Bearing these examples in mind, we clarify that the knowledge gap we address refers to the interplay between CCA investments specifically. Moreover, in the Discussion, we discuss the potential of our finding to broaden the uptake of public policies that already enable further private CCA (using the new references in the Introduction as examples, Lines 393-394). We discuss how the crowding-in mechanism, combined with CCA effectiveness assessments to prevent maladaptation, can help private firms achieve a sufficient level of adaptation where they can focus on investments that grow businesses and create jobs (Lines 459-462). Finally, given the resolution and temporal span of our data, we call for more detailed analysis of this mechanism through e.g. “in-depth interviews with individual businesses revealing their motivation for CCA investments” (Lines 464-467).

Detailed comments by section:

1. Title: Usually the term “spending” is used in the context of public expenditure, so I suggest using “adaptation investment” or the like instead.

Response: Thank you for pointing out this important distinction and suggesting a suitable alternative. We have implemented your suggested change in the title, and throughout the paper. Moreover, we have defined adaptation investment as including all capital

expenditures on CCA (Lines 104-105). For the sake of readability, we refer to adaptation investments and adaptation expenditures interchangeably in the text, as clarified in Lines 103-104. We hope these changes make the text more accessible to readers across a variety of academic fields.

2. Abstract: What's the difference between welfare and well-being? More clarity is needed in the abstract. Also, it is not clear what the "adaptation economy" is. I suggest to not use this term.

Response: Thank you for this comment, and we apologise if the abstract lacked clarity. We have now removed references to welfare and well-being, as they were not present elsewhere in the paper, and now refer to "an increasing need for climate adaptation measures to protect both the population and the economy" (Lines 14-15).

Regarding the use of the term 'adaptation economy', we have followed your advice and removed it from the manuscript. Upon reflection, it is more appropriate when describing transactions between firms to provide adaptation services (i.e., who buys what AND from whom), yet the focus of our manuscript excludes the latter. Where relevant, we simply refer to 'adaptation investments or expenditures' (as defined in Lines 104-105) and avoid this umbrella term.

Introduction

3. Additional to source 3) more references are needed. At least the IPCC should be cited, but ideally a bunch of original research papers.

Response: Thank you for the comment. Regarding the IPCC citation, we have now changed our reference from the Summary for Policymakers to the Synthesis Report (Ref 2) and included it following the indicated statement. Additionally, we have included two original research papers supporting this statement; these articles are part of the climate attribution literature (Line 32).

4. second para is lengthy. I suggest splitting it up.

Response: Thank you for pointing out this issue with readability. We have now split the paragraph at the point in which we introduce the understudy of private CCA: "Nevertheless, while public CCA policies have been extensively researched and are increasingly included in macroeconomic assessments^{11,16,19,20}, private CCA is comparatively understudied" (Line 54).

5. Line 77: public sector not only "delivers" (I think what the authors mean is "supply") infrastructure but also protects it.

Response: We thank Reviewer #2 for noticing this point. In Lines 84-85, we now stress that "CCA planning has relied on the public sector to **supply and maintain** critical infrastructure (...)".

6. Lines 81-85: the social/equity dimension of CCA is important to raise here. The question is also whom the government protects and how to use scarce resources from that perspective.

Response: We agree that the social/equity dimension of CCA is important. The revised manuscript now eludes to this point by highlighting that public resources are often allocated through benefit-cost maximisation, which can prioritise high value assets owned by the rich (Lines 96-98).

7. Line 99ff: are foreign direct investments included in the sectoral investments?

Response: We appreciate the opportunity to clarify this point, as it was missing in the previous version of the manuscript. In Lines 104-105, we now state “These include all capital expenditures on CCA, including those funded through foreign direct investments (...)”.

8. The wording of the “adaptation economy” that is “growing steadily” sounds very positive (à la increases GDP), but I think this is not the message that the authors want to deliver. It is important to stress that adaptation investment is a kind of cost to the macroeconomy, since the resource could be used elsewhere more productively. Very often the only return on investment in adaptation is the avoided or reduced damages.

Response: Thank you for bringing up this important point for discussion. We agree that adaptation investment growth is not positive per se, as these investments need to be efficient and serve a damage-mitigating purpose. Thus, we have followed your suggestion of toning down the positivity associated with “growing adaptation”, by referring to increases or rises instead (Line 134-139), and point out the importance of assessing adaptation effectiveness in the Discussion, also in the context of maladaptation (Line 462-464). Finally, we explicitly mention that efficiently achieving a sufficient level of adaptation would allow firms to focus on traditional investments (i.e., growing the business, create jobs) (Lines 467-470).

Results

9. line 113: it is 243% not 143%

Response: Thank you for noticing this typo, we agree and have implemented the change in Line 143.

10. line 116: “level” should be “share”

Response: Thanks. We have changed this in Line 135 and where relevant elsewhere in the manuscript.

11. The first paragraph ends with a lot of speculation. A more profound reference basis is needed for many of the claims (e.g. ref 43 is certainly not enough, also because it is very outdated).

Response: We agree with your suggestion to add more recent multi-regional assessments to support these claims. Thus, we have removed the old ref 43 and added three more recent assessments covering a variety of hazards and sectors across Europe. Additionally, we have removed speculation regarding the growth of the ratio itself and the explicit ability of certain countries to fund CCA in the future. Instead, we highlight increasing adaptation needs and greater financing burdens in countries with higher climate damages and lower economic productivity (Lines 152-156).

12. lines 128ff: Having a higher adaptation level in place is eventually the same as being relatively less impacted.

Response: Thank you for making this point. We have re-edited this sentence to focus exclusively on impact, as a point on adaptation is made in the following sentence. It now reads as “Conversely, the lower half of the ranking is populated by smaller and Northern European states, which are projected to be relatively less impacted by climate shocks” (Line 150-152).

13. Much of the results could also be explained by income differences. A low share of GDP of a rich country might be sufficient for the country’s adaptation requirements, whereas a high share of GDP of a lower income country might not. This dimension is currently underdeveloped.

Response: Thank you for pointing out this potential interaction. We agree with Reviewer #2 that a higher national income could lead to lower adaptation investment per unit of GDP, assuming a relatively even physical requirement of infrastructure or labour to protect. However, both the Netherlands and Germany are within the top 8 countries in the ranking, so for at least these two cases, a higher national income per capita is not diminishing how much is proportionally spent in adaptation. Overall, we think it is important to highlight the relative clustering of lower income states towards the top of the ranking, pointing this out as a potential problem in the future as investments in adaptation have an opportunity cost over other development endeavours (in line with Comment #8). Therefore, when discussing the lower income countries at the top of the ranking, we now clarify: “Given their comparatively lower levels of income, similar adaptation measures might represent higher relative costs per unit of GDP” (Line 149-150).

14. Fig. 1. Isn’t Denmark also heavily affected by sea level rise, so this same as for the Netherlands.

Response: Indeed, it is true that both the Netherlands and Denmark are under significant threat from sea-level rise this century. However, comparatively, the Netherlands is at significant risk of river flooding and compound flooding events, which are expected to become a greater threat in the short term. Note that the category “Flooding” includes adaptation to all kinds of floods (coastal, pluvial and fluvial). Moreover, in the hazard-specific breakdown, we can see that the percentage of investment in flooding is much higher for the Netherlands (36.8%) than for Denmark (26.7%).

To address your point, the revised manuscript now clarifies: “As expected, given its historic exposure to both river and coastal flooding⁴⁷, and high potential for compound hazards, the Netherlands spends the most on adaptation relative to its economy (...)” (Lines 142-143).

15. lines 150ff: I strongly recommend to not use the terms “underinvestment” and “overinvestments” in this context. In fact, what the analysis shows is simply a ratio, but this does not say anything about the requirement of a sector’s adaptation. For some sectors a very low share might be perfectly fine, whereas for some others an already high share might still be insufficient. What would be needed is a spatial mapping of sectoral activity and an overlay with hazards to then derive the risk and adaptation requirements by sector. In fact, it can never happen – by construction – that all cells in Table 1 turn green.

Response: We would like to thank Reviewer #2 for the recommendation and elaboration on their argument. Indeed, a spatial mapping of sectoral activity overlaid with hazard maps (if we take flooding as an example) would be ideal. Honestly speaking, our first attempt at constructing this metric involved developing a country-sector specific proxy for risk to understand objective needs for each country-sector combination. When preparing the initial submission, we have studied in detail different databases reporting damages from the types of hazards that constitute the focus of our analysis. Unfortunately, the data gathered on direct damages for past events and future projections is sparse (across countries and largely unavailable at the sectoral level) and inconsistent, making it difficult to combine, even for Europe. Therefore, we believe it is a great avenue for future work, where the analysis can be restricted to fewer country-sector combinations before upscaling it to a multi-regional, multi-sectoral assessment. We have further highlighted this point in the Discussion (Lines 485-489, 492-494).

Regarding the use of the terms “under-” and “overinvestment”, we agree with Reviewer #2 that the ratio provides no information regarding the requirement of a sector’s adaptation. Additionally, these terms might mean different things for readers of different disciplinary backgrounds. Therefore, to highlight the macroeconomic nature of our piece, we now introduce the concept of adaptation investment intensity, and refer to “intensive” adaptation for sectors in blue, and “non-intensive” for sectors in red, similar to how certain sectors are referred to as capital- or labour-intensive in the broader macroeconomic literature (Lines 201, 207, 226 and throughout the article). By doing this, we remove any overinterpretation regarding the suitability of the level of adaptation, while still illustrating the clear sectoral trends. The change in the colour palette further reflects the idea that a high ratio need not necessarily imply a “good” level of adaptation, which would be hinted at by a red-green scale. Finally, we point out in Table 1 that by construction all rows must have blue and red cells, as the ratios are defined around a national weighted average equal to 1.0 (Table 1, Lines 245-247).

16. The “Real estate” sector needs more care. What does this sector really show? Is it only the service provision, or does it really include all the assets (buildings)?

Response: Thank you for bringing up this very important distinction. The Real Estate Services sector (Sector L) in the adaptation dataset refers to expenditures related to

adaptation by businesses, which include the offices of realtors and real estate agencies, but also the existing assets of housing corporations. Notably, our dataset does not include private household adaptation measures as implemented by private individuals, neither in owner-occupied dwellings nor in tenant-occupied homes (owned by non-corporate entities). Given that the GVA estimate for Sector L in Eurostat includes both imputed rents (i.e., an estimate of how much a house owner would pay in rent if they rented the dwelling to themselves) and rental income by private landlords, we cannot reliably compare the GVA share with the adaptation share without household adaptation spending data. Given the article's focus on CCA by businesses, we believe that assessing the level of adaptation of the real estate sector including private household adaptation would require its own specific analysis, as is already commonly done in the adaptation literature. Therefore, we have decided to not include the estimate in our table and have provided this justification in the Methods when describing the descriptive statistics (Lines 591-599).

Specifically, we have removed all values related to Sector L from Table 1 (we have kept the column and made it grey for transparency) and have naturally removed the previous discussion related to real estate. Additionally, we have re-calculated all adaptation intensity ratios after removing the contribution of Sector L to both GDP and total adaptation (in line with the point made by Reviewer #2 in Comment #15). The new estimates are marginally different to those in the previous manuscript, but these changes are not large enough to alter the qualitative interpretation of the results for other sectors. Finally, as it pertains to the econometric analysis, we have assumed that the GVA growth rate for Sector L (which is used as an explanatory variable) is representative of the growth rate in GVA exclusively coming from real estate companies and housing corporations (this is also disclosed in Lines 645-647). We believe that this assumption is defensible on the grounds that higher home prices normally result on higher profits for real estate companies and housing corporations. Additionally, the dimensional jackknife analysis shows that removing any individual sector does not significantly change the regression results (as reported in the Supplementary Information, p.24).

17. Sector "Water supply". Note that this sector is closely connected to flood protection infrastructure.

Response: Thank you for the comment, we have included this point now in the text when discussing the 'Water Supply' sector (Lines 202-204).

18. Line 173 "may signal" – this is speculative

Response: We agree that this sentence is too speculative for the Results section. In the revised manuscript, we no longer speculate about future damages and have added a reference reviewing the mining sector's vulnerability to climate change (Lines 209-212).

19. Line 207 "per hazard" is confusing. It sounds like the authors divided by the number of events, which is not the case, I assume (rather it is the attribution of an adaptation investment to a specific hazard type).

Response: Thank you for highlighting that the current wording is unclear. We now refer to "hazard type" to avoid further confusion (Line 260).

20. Lines 209ff: I would not use the term “attract”, since no causality it proven. It simply gives an approximation of the main risks at the moment. The key question to answer would be whether an increase in risks really leads to increased investment in adaptation, but this would require a longer time series. But maybe the dataset allows to look into extreme events and check whether investments have increase in the subsequent years (for specific country-sector-hazard combinations).

Response: Thank you for pointing out that the sentence read as inferring causality, which was not our intention. We simply wanted to indicate the presence of this cross-country trend to the reader, and as such we have now revised the sentence to “firms invest proportionately the most in adaptation to Heatwaves” (Line 263).

21. Lines 249ff: This requires some cross-checks to other studies.

Response: Thank you for the comment. We agree that cross-study comparisons on the relative share of public relative to private CCA would be desirable. However, we have not found any studies that specifically capture this relationship at macroeconomic scale, partially due to the fragmentation of the literature between public and private action. Specifically, we have done a thorough literature search before starting the study, repeated it during the drafting period, and have re-run it again during this revision stage, but did not find any literature that reports the relative share of public adaptation (under any specific definition) relative to private adaptation. We would very much appreciate it if Reviewer #2 could suggest any specific literature references which we might have missed.

For now, in the absence of alternatives, we have compared the scale of public adaptation specifically to that of another recent study, which is roughly 2 times higher in our dataset (for the three countries considered in the other study: The Netherlands, Spain, Austria) (now discussed in Lines 418-421). We believe this difference stems from the inclusion of heatwave adaptation in our dataset (not included in the other study). Given that we cannot provide a real comparison on the share presented in the Results, we highlight this point in the discussion when discussing data limitations and future data needs (Lines 491-492).

22. On the interplay between public-private: This is very interesting, but deserves at least mor discussion. What about public-private-partnerships, for example?

Response: Thank you for bringing up this important consideration that was missing from the previous version of the manuscript. PPPs are important frameworks for local implementation of adaptation measures, particularly under major budgetary constraints. We now briefly discuss their potential in the Introduction (Lines 92-93) and the Results (Line 270-273).

As it relates to the interplay between public and private investments, we have provided additional examples of existing synergies between public and private adaptation measures and have framed the knowledge gap in the Introduction more specifically around the interplay between CCA investments (Lines 84-85). We return to this point in the Discussion, pointing out that while our findings cannot provide explicit recommendations

on how to effectively direct public spending in CCA, they can “act as a starting point to broaden the uptake of public policies that already enable further private CCA” (Lines 323-324).

23. Lines 289ff: The slow-down in investments might be due to the fact that adaptation happens in waves; and often only after an extreme event. This needs at least discussion.

Response: Thank you for pointing out this potential mechanism. We agree that the occurrence of extreme events may lead to spikes in adaptation uptake, which is why we have included the sum of recorded extreme events in the country in the previous 5 years in the specification. We now point out the existence of uptake spikes as further evidence for the need to include this variable in the text (Line 332). We are aware that the hazard count in the previous five years is an imperfect variable to address this mechanism, which is why we wanted to include further event-specific data in our regression analysis (i.e., monetary losses, population affected, etc.). Unfortunately, data beyond the occurrence of the event itself is sparse and hard to compare across countries, especially as some recorded events do not include economic (or other) losses in the Risk Data Hub database. Therefore, we call for more systematic reporting of damages as an area for future work in the Discussion (Lines 492-494), to improve our ability to attribute spikes in adaptation uptake to recent events (and the magnitude of those events).

24. Line 296: How is population density included? National average?

Response: Thank you for the question, and apologies for not clarifying this earlier. Indeed, the population density included in the analysis is the national average, which we have now reported in the Results (Line 330). Our reasoning for its inclusion is that countries with higher average population density are likely to overall spend more in adaptation to protect large climate-sensitive urban areas, as discussed in the economic agglomeration literature. We have now clarified this further in the Methods as well (Lines 655-656).

25. Line 313: “should be considered” – say how!

Response: Thank you for pointing out the lack of clarity in this sentence. We now explicitly mention the allocation of public adaptation resources and the design of adaptation uptake policies (Lines 389-390).

Discussion

26. Line 360: “underinvestment” is not measured properly, so an increase based on this index might be inefficient (see my comment on that above). C-B ratios would be needed, ideally including economy-wide effects.

Response: Thank you for this suggestion. After removing the under- and overinvestment interpretations throughout the paper, we have updated this section in the Discussion as well. In Lines 445-447, we now encourage policymakers to consider sectoral adaptation intensities and their criticality to the national economy when allocating adaptation resources, on top of each sector’s exposure to climate-induced hazards. Additionally, we point to the use of benefit-cost ratios in the allocation of these resources.

27. I expect a thorough update of the discussion section after the revisions.

Response: We have followed your advice and made significant edits in the manuscript. The Discussion section has now been updated to reflect the changes to the sectoral analysis (Line 427-432), where adaptation intensity is discussed, and to highlight the implications of the crowding-in mechanism as a means to more efficiently achieve sufficient private CCA (Line 467-470). Additional changes on future work regarding data needs on exposure to extreme weather events and other studies that can validate our results have also been added (Line 491-494).

Methods

28. As given as a major point of concern above, a detailed description of how the dataset was constructed is needed.

Response: We have addressed this comment above under Major Issues, Comment #1.

Response to Reviewers – Private investments in climate change adaptation are increasing in Europe, although sectoral differences remain

Reviewer #1 (Remarks to the Author):

The authors have carefully considered the reviewers' comments and presented the changes transparently. I am satisfied with the changes, which have significantly improved the manuscript, particularly the discussion of the results. Therefore, I have no further comments and recommend publication.

On behalf of the entire team, we would like to thank Reviewer #1 for the time taken to review our work and for their constructive feedback, which has helped to significantly improve our manuscript. Thank you!

Reviewer #2 (Remarks to the Author):

The authors have taken up all of my concerns and have addressed them satisfactorily.

On behalf of the entire team, we would like to thank Reviewer #2 for their appreciation of our work and for the time dedicated to improve the quality of the manuscript. Their constructive and detail-oriented approach has been very helpful in framing and discussing our results, as well as in increasing the clarity of our piece's message. We are happy that our efforts have addressed their concerns satisfactorily. Thank you!

I only have some very minor points, which I'd like to see resolved, before publishing.

In Table 3 is SI, some entries in the "DATA sources" column only are labelled with a number. Why? What does that mean? Are they confidential?

Thank you for pointing this out, the phrasing in the caption was a bit unclear. Indeed, the data sources labelled with a number are confidential. We now clarify in the caption that "Some data sources are confidential, and are thus labelled with a number in the "DATA Source" column instead of the source's name" (Supplementary Table 3, Page 8, SI).

The sentence: "protect both the population and the economy" – I think this is too narrow. What about ecological systems, biodiversity etc.? Please rephrase.

Thank you for your comment. We agree that adaptation measures have a wider scope than to protect the population and the economy. Thus, and given the word count constraints in the abstract we instead refer to "protect people, nature and the economy" in the new manuscript.

At some points the term "firms" is in fact wrongly used, as the analysis focuses on economic sectors. Of course, "firms" can be used in the arguments etc., but not when e.g. saying what you present. For example in the discussion: "our study provides an extensive overview of the current state of private CCA investment by firms...", which is not true.

Thank you for pointing us to this important distinction in the paper. When presenting our results, we now refer to “sectors” or “private-sector CCA” in the results and early discussion (e.g., “Our study provides an extensive overview of the current state of private-sector CCA investment...”(Discussion, Page 8)). As suggested, we have kept the term “firms” in the introduction and in argumentation in the discussion.

Concerning the low intensities in sector A: could it be that the public sector steps in here a lot and protects this sector (via flood protection for example)? What about subsidies for adaptation (e.g. for irrigation infrastructure)? Is this included? (Sorry for not having brought this up in the initial review). Maybe a sentence or two can be added to the manuscript?

Thank you for the insightful comment. We agree that further reflection on the agricultural sector is of great importance when discussing private-sector adaptation. In our study, the underlying dataset does not identify whether the investments made by the agricultural sector are a result of subsidies or not. As a consequence, all expenditure in agricultural adaptation measures are counted as part of sector A. In the previous version of the manuscript, we had clarified this for all sectors in the section “The interplay between public and private adaptation” as it is relevant for the econometric results.

However, given its particular importance for agriculture, we have now revised the manuscript to point it out when discussing this sector in Lines 183-186. In particular, we add that

“Given the importance of subsidies to European agriculture⁵⁴ and that all agricultural adaptation measures – including those financed through subsidies – are part of the sector’s CCA expenditures (see Supplementary Table 4), government policies can play a key role in mitigating these differences in adaptation intensity.”